# Making Large Language Models Better Reasoners with Alignment

## Abstract

Reasoning is a cognitive process of using evidence to reach a sound conclusion. The reasoning capability is essential for large language models (LLMs) to serve as the brain of the artificial general intelligence agent. Recent studies reveal that fine-tuning LLMs on data with the chain of thought (COT) reasoning process can significantly enhance their reasoning capabilities. However, we find that the fine-tuned LLMs suffer from an *Assessment Misalignment* problem, i.e., they frequently assign higher scores to subpar COTs, leading to potential limitations in their reasoning abilities. In this paper, we introduce an *Alignment Fine-Tuning (AFT)* paradigm with a novel *Constrained Alignment Loss* to alleviate the assessment misalignment problem. Specifically, the proposed loss has two objectives: a) Alignment, which guarantees the scores of high-quality COTs surpass that of subpar ones; b) Constraint, which keeps the subpar scores confined to a reasonable range to prevent the model degradation. Extensive experiments on four reasoning benchmarks with both binary and ranking feedback demonstrate the effectiveness of AFT. AFT also performs well in multi-task and out-of-distribution situations. Furthermore, we also delve deeply into recent ranking-based alignment methods, such as DPO, RRHF, and PRO, and discover that the constraint, which has been overlooked by these approaches, is also crucial for their performance.

## 1 Introduction

Reasoning is a cognitive process that involves utilizing evidence to reach a well-founded conclusion (Qiao et al., 2023; Huang & Chang, 2023). Recently, there has been a growing focus on enhancing the reasoning abilities of Large Language Models (LLMs) (Li et al., 2023b; Yuan et al., 2023a; Luo et al., 2023; Mukherjee et al., 2023), because LLMs still lack reasoning skills (Wang et al., 2023b;d; Zheng et al., 2023; Kaddour et al., 2023; Zhao et al., 2023a) that are essential for them to serve as the brain of artificial general intelligence agents (Wang et al., 2023a; Yao et al., 2023; Song et al., 2023b).

Recent works (Chung et al., 2022; Hsieh et al., 2023; Mukherjee et al., 2023; Yue et al., 2023) find that training LLMs using data with a chain of thought (COT) reasoning process is a very effective method to improve the reasoning ability of LLMs. These studies typically train LLMs using maximum likelihood estimation (MLE), and employ a next-token prediction objective. However, MLE only assigns probability mass to the reference COT, which contradicts reasoning tasks where various reasoning paths can lead to the correct answer. In this paper, we find that such vanilla fine-tuning (VFT) paradigm causes LLMs to suffer from an *Assessment Misalignment* problem, i.e., LLMs struggle with accessing the quality of different COTs, ultimately limiting their reasoning capabilities. Take Figure 1 as an example, VFT-LLMs learn to generate the *Reference Answer* for the given *Question* by allocating probability mass to this *Reference Answer* and treating all other answers as negative outcomes. As a result, they struggle to assess the quality of other answers and tend to assign lower perplexity (higher score) to *incorrect Candidate Answer 1* compared to the *correct Candidate Answers 2*.

This behavior of VFT-LLMs is not consistent with that of humans, as humans have the ability to access the quality of different COTs after learning to reason. In addition, our pilot experiments (Section 3) find that after the same VFT process, the LLMs with better reasoning performance can give a more reasonable assessment to different COTs. Therefore, we hypothesize that we can improve the reasoning ability of LLMs by alleviating the assessment misalignment problem caused by VFT.

| | |
|---|---|
| **Question:** Weng earns $12 an hour for babysitting. Yesterday, she just did 50 minutes of babysitting. How much did she earn? | **Candidate Answer 1 (PPL 1.90) :** Weng earns 12/60 = <<12/60=0.2>>0.2$ per minute Yesterday she did 50 minutes of babysitting, so she earned 50 * 0.2 = <<50*0.2=10>>10$. ✓ |
| **Reference Answer (PPL 1.05) :** Weng earns 12/60 = $<<12/60=0.2>>0.2 per minute. Working 50 minutes, she earned 0.2 x 50 = $<<0.2*50=10>>10. | **Candidate Answer 2 (PPL 1.35) :** Weng earns $12/hour x 60 minutes = $<<12*60=720>>720 per hour. Working 50 minutes, she earned $720 x 50/60 = $<<720*50/60=40>>40. ✗ |

Figure 1: Perplexity of different answers given by the vanilla fine-tuning (VFT) LLM, where LLM assigns a lower perplexity to the incorrect candidate answer compared to the correct candidate answer.

To address the assessment misalignment problem, in this paper, we propose an alignment fine-tuning (AFT) paradigm to improve LLM reasoning with three steps: **1)** fine-tuning LLMs using COT training data; **2)** generating multiple COT responses for each question using the fine-tuned LLMs, and categorizing them as positive and negative based on whether they deduce the correct answer; **3)** calibrating the scores of positive and negative responses given by LLMs with a novel Constrained Alignment (CA) loss. Specifically, the CA loss ensures that all positive scores (the scores of positive COTs) are larger than negative scores. In addition, the negative scores are protected by a constraint term, which is proven to be very important in preventing model degradation. Beyond just binary positive and negative feedback, the CA loss can be seamlessly adapted to ranking situations when ranking feedback is accessible. Furthermore, we also delve deeply into recent ranking-based methods for alignment, such as DPO (Rafailov et al., 2023), PRO (Song et al., 2023a) and RRHF (Yuan et al., 2023b), and find that the constraint, which has been overlooked by these approaches, is also crucial for their effectiveness.

In summary, our contributions are: **1)** We discover that LLMs fine-tuned by the vanilla fine-tuning (VFT) paradigm suffer from an Assessment Misalignment problem: they frequently assign lower scores to high-quality COTs compared to low-quality ones, which hinders their reasoning ability; **2)** We present an alignment fine-tuning (AFT) paradigm with a novel constrained alignment loss to address the identified problem; **3)** Experiments on four reasoning benchmarks with both binary and ranking feedback demonstrate the effectiveness of AFT. AFT also performs well in multi-task and out-of-distribution situations; **4)** We delve deeply into recent ranking-based alignment methods and find that the constraint, which has been overlooked by them, is also crucial for their performance.

## 2 RELATED WORKS

### 2.1 IMPROVE REASONING OF LARGE LANGUAGE MODELS

Reasoning is a cognitive process that utilizes evidence to reach a well-founded conclusion and is a core ability of LLMs to serve as the brain of the artificial general intelligence agent. Researchers have proposed a lot of methods to improve the reasoning ability of LLMs, which can be broadly divided into three groups: 1) *pre-training*: The pre-training methods pre-train LLMs on a vast of unsupervised datasets, such as the pile (Gao et al., 2020), the stack (Kocetkov et al., 2022), with a simple next token prediction objective. Researchers find that a larger model pre-trained on more data tends to have better reasoning ability (OpenAI, 2023; Anil et al., 2023; Touvron et al., 2023); 2) *fine-tuning*: The fine-tuning methods can also enhance the reasoning ability of LLMs. Researchers have found that fine-tuning LLMs on the data with the reasoning chain-of-thought process can significantly improve the reasoning of LLMs (Mukherjee et al., 2023; Chung et al., 2022; Li et al., 2023a); 3) *prompting*: The prompting methods aim to improve the reasoning ability of LLMs by carefully designed prompting strategy without updating the model parameters (Wei et al., 2022; Wang et al., 2023c; Bi et al., 2023), which is very convenient and practical. In this paper, we focus on the fine-tuning methods and find that traditional vanilla chain-of-thought fine-tuned LLMs suffer from an assessment misalignment problem, which hinders their reasoning ability. To this end, we propose an alignment fine-tuning paradigm to address this problem to enhance the reasoning ability of LLMs.

### 2.2 PREFERENCE ALIGNMENT OF LARGE LANGUAGE MODELS

Preference alignment research focuses on directing AI systems toward human-intended preferences Wang et al. (2023e). There are three primary categories of preference alignment methods: 1) *Reinforcement Learning from Human Feedback (RLHF)* (Ouyang et al., 2022; Wu et al., 2023), which

trains a reward model by utilizing human feedback. The reward model acts as a reward function for optimizing an agent's policy through reinforcement learning (RL) techniques, such as Proximal Policy Optimization (Schulman et al., 2017). RLHF is employed to align powerful LLMs, like ChatGPT and GPT-4. However, RL-based methods face limitations concerning training efficiency and complexity; 2) *Rejective Sampling* (Touvron et al., 2023; Gulcehre et al., 2023; Dong et al., 2023; Liu et al., 2023), which aligns large language models by sampling multiple responses and choosing high-quality ones to further enhance their training; 3) *Supervised Fine-tuning with Ranking* (Liu et al., 2022; Yuan et al., 2023b; Zhao et al., 2023b; Song et al., 2023a; Rafailov et al., 2023), which incorporates a ranking loss to help LLMs align with human preferences. Previous preference alignment research has mainly focused on improving the safety of LLMs, frequently neglecting the importance of preference alignment for reasoning. In this paper, we point out the effectiveness of preference alignment for reasoning and introduce a novel constrained alignment loss to make LLMs better reasoners.

## 3 PILOT EXPERIMENTS

In this section, we first briefly introduce the vanilla fine-tuning (VFT) paradigm, and then we demonstrate the *Assessment Misalignment* problem of VFT for reasoning.

### 3.1 VANILLA FINE-TUNING

VFT finetunes LLMs on a dataset $\{(q_i, c_i, a_i)\}_{i=1}^N$ with $N$ examples. Each example consists of a question $q_i$, a COT reasoning process $c_i$, and an answer $a_i$. The LLMs are finetuned to generate the reference response $r_i = [c_i; a_i]$ based on $q_i$ with a MLE objective loss function:

$$\mathcal{L}_{VFT} = -\sum_{j=1}^{|r_i|} \log P(r_{i,j} \mid r_{i,<j}, q_i; \theta). \tag{1}$$

where $\theta$ is the model parameter and $r_{i,j}$ is the $j$-th token of $r_i$.

### 3.2 ASSESSMENT MISALIGNMENT OF VFT FOR REASONING

Intuitively, the MLE objective seeks to exclusively allocate probability mass to the reference COT $c_i$ for question $q_i$, which does not correspond with the characteristics of reasoning tasks, where the correct COT is not limited to the reference one. This objective uniformly treats all other correct and incorrect COTs as negative examples. As a result, it will impede LLMs from learning to assess the quality of various COTs and degrade their reasoning ability.

To demonstrate this, we first fine-tune LLama-7B, LLama-13B, LLama2-7B, and LLama2-13B on the training data of GSM8k and ECQA with Equation 1 (please refer to Section 5.1 for the detailed VFT settings). Then, for each question $q_i$ in the training data, we use VFT-LLMs to generate three positive COTs $\{c_i^{p1}, c_i^{p2}, c_i^{p3}\}$ that induce to the correct answer and three negative COTs $\{c_i^{n1}, c_i^{n2}, c_i^{n3}\}$ that induce to the incorrect answer, respectively. Upon manually examining 50 examples, we observe that the quality of positive COTs is noticeably better than that of negative COTs.

We further compute the token-averaged log-likelihood score of each positive and negative COT $c$ using the fine-tuned LLMs as follows:

$$s_\theta^c = \frac{1}{|c|}\sum_{j=1}^{|c|} \log P\left(c_j \mid c_{<j}, q; \theta\right), \tag{2}$$

where $q$ is the corresponding question. It is reasonable to expect that the fine-tuned LLMs will be able to assess the quality of different candidate COTs of previously encountered questions, i.e., assigning higher scores to the positive ones. Therefore, we use an assessment accuracy $\mathbf{A_{Accuracy}}$ to assess the capability of fine-tuned LLMs in assigning appropriate scores to various COTs:

$$\mathbf{A_{Accuracy}} = \frac{1}{9N}\sum_{i=1}^{N}\sum_{j=1}^{3}\sum_{k=1}^{3} \mathbb{I}(s_\theta^{c_i^{pj}} > s_\theta^{c_i^{nk}}) \tag{3}$$

As shown in Table 1, the assessment accuracy of the VFT-LLMs falls short of expectations, with an average $\mathbf{A_{Accuracy}}$ of merely around $70\%$ on GSM8K and $62\%$ on ECQA, respectively. Note that

| MODELS | GSM8K (PEARSON = 0.93) | | ECQA (PEARSON = 0.98) | |
|--------|------------------------|------------------------|------------------------|------------------------|
| | $\mathbf{T_{Accuracy}}(\%)$ | $\mathbf{A_{Accuracy}}(\%)$ | $\mathbf{T_{Accuracy}}(\%)$ | $\mathbf{A_{Accuracy}}(\%)$ |
| LLama-7B | $36.48\pm0.92$ | $68.41\pm0.32$ | $70.40\pm0.92$ | $61.62\pm0.01$ |
| LLama2-7B | $40.71\pm0.16$ | $71.22\pm0.12$ | $72.34\pm0.22$ | $61.96\pm0.02$ |
| LLama-13B | $42.07\pm0.15$ | $72.25\pm0.23$ | $72.74\pm0.43$ | $61.89\pm0.01$ |
| LLama2-13B | $47.29\pm1.24$ | $73.06\pm0.78$ | $74.76\pm0.56$ | $62.29\pm0.01$ |

Table 1: The final task accuracy ($\mathbf{T_{Accuracy}}$) and the assessment accuracy ($\mathbf{A_{Accuracy}}$) of different vanilla fine-tuned models. $\mathbf{T_{Accuracy}}$ and $\mathbf{A_{Accuracy}}$ exhibit a strong positive correlation, with Pearson Correlation Coefficients of 0.93 and 0.98 at GSM8K and ECQA, respectively.

this is a two-class classification problem where a random baseline can achieve the $50.00\%$ accuracy. These results show that the assessment ability of VFT-LLMs is far from expected, as they cannot accurately discern the quality of various COTs of previously learned questions. This behavior of VFT-LLMs is not consistent with that of humans, as humans have the ability to access the quality of different COTs after learning to reason. In addition, we also notice that LLMs with stronger reasoning abilities have better assessment accuracy. Specifically, the task accuracy and the assessment accuracy exhibit a strong positive correlation, with Pearson Correlation Coefficients of 0.93 and 0.98 at GSM8K and ECQA, respectively. This observation inspires us to improve the reasoning ability of LLMs by aligning their scoring behaviors with the golden standard assessment.

## 4 METHODOLOGY

We have demonstrated that the scoring behaviors of vanilla fine-tuned LLMs exhibit misalignment with the gold standard assessment. In this section, we propose an alignment fine-tuning (AFT) paradigm to address this problem to enhance their reasoning ability. Specifically, on top the VFT objective $\mathcal{L}_{VFT}$, AFT further introduce an alignment objective $\mathcal{L}_A^*$:

$$\mathcal{L}_{AFT} = \mathcal{L}_{VFT} + \mathcal{L}_A^*. \tag{4}$$

In the following part of this section, we will introduce the design process of $\mathcal{L}_A^*$.

### 4.1 GENERATE CANDIDATE COTS FOR TRAINING DATA

To implement AFT, we first need to generate multiple COTs for each question in the training set. For each training example $(q, c, a)$, we first sample $k$ generation results $\{(c_i, a_i)\}_{i=1}^k$ from the VFT-LLMs based on the input question $q$. Then, we divide these generation results into two groups, namely positive group $\mathbf{G}_P$ and negative group $\mathbf{G}_N$, based on the correctness of their answer. Formally, a generation results $(c_i, a_i)$ belongs to $\mathbf{G_P}$ if $a_i = a$, otherwise it is part of $\mathbf{G}_N$. Generally, the quality of COTs in the positive group $\mathbf{G}_P$ is better than that of $\mathbf{G}_N$.

### 4.2 ALIGNMENT

As demonstrated by our pilot experiment, VFT-LLMs fail to give reasonable scores to COTs in $\mathbf{G_P}$ and $\mathbf{G_N}$. To align the scoring behaviors of LLMs with the golden standard assessment, we need to design an objective to let the scores of all positive COTs in $\mathbf{G_P}$ larger than that of negative COTs in $\mathbf{G_N}$. This objective bears resemblance to contrastive learning, which aims to ensure that the score of positive example is larger than those of all negative examples, utilizing an InfoNCE loss:

$$\mathcal{L}_{InfoNCE} = -\log\left[\frac{\exp(s_\theta^{c_p})}{\exp(s_\theta^{c_p}) + \sum_{c_n \in \mathbf{G_N}} \exp(s_\theta^{c_n})}\right] = \log\left[1 + \sum_{c_n \in \mathbf{G_N}} \exp(s_\theta^{c_n} - s_\theta^{c_p})\right] \tag{5}$$

Intuitively, minimizing Equation 5 aims to make the positive score $s_\theta^{c_p}$ larger than all negative scores. However, since there is more than one positive example in $\mathbf{G}_P$, inspired by (Su et al., 2022; Wang et al., 2022), we extend $\mathcal{L}_{InfoNCE}$ to accommodate multiple positive examples:

$$\mathcal{L}_A = \log\left[1 + \sum_{c_p \in \mathbf{G_P}} \sum_{c_n \in \mathbf{G_N}} \underbrace{\exp(s_\theta^{c_n} - s_\theta^{c_p})}_{\text{alignment term}}\right] \tag{6}$$

where $s_\theta^c$ is the average log-likelyhood score of the COT $c$ calculated by Equation 2. Minimizing $\mathcal{L}_A$ encourages all positive scores to be larger than all negative scores.

## 4.3 CONSTRAINT

Nevertheless, although the quality of negative COTs may not be as high as that of positive COTs, they still retain a respectable quality, as they are sampled from fine-tuned and powerful LLMs. We find that reducing their scores by Equation 6 without setting any constraint will result in the degradation of the LLMs. Therefore, we further design two constrained methods, Detached Constraint (DC), and Boundary Constraint (BC) to avoid such catastrophe.

### 4.3.1 DETACHED CONSTRAINT

To prevent model degradation, DC adds constraint to negative scores by detaching their gradient:

$$\mathcal{L}_A^{DC} = \log \left[ 1 + \sum_{c_p \in \mathbf{G_P}} \sum_{c_n \in \mathbf{G_N}} \underbrace{\exp\left(\mathbf{D}(s_\theta^{c_n}) - s_\theta^{c_p}\right)}_{\text{detached alignment term}} \right], \tag{7}$$

where $\mathbf{D}(\cdot)$ denotes the detach operation, which means the gradient would not back-prop through the negative scores. As a result, $\mathcal{L}_A^{DC}$ achieves the alignment by only increasing positive scores without explicitly decreasing negative ones.

### 4.3.2 BOUNDARY CONSTRAINT

Besides DC, we also want to explore whether better results can be obtained by marginally explicitly decreasing negative scores. To this end, we propose BC that adds a constraint term to $\mathcal{L}_A$:

$$\mathcal{L}_A^{BC} = \log \left\{ 1 + \sum_{c_p \in \mathbf{G_P}} \sum_{c_n \in \mathbf{G_N}} \left[ \underbrace{\exp(s_\theta^{c_n} - s_\theta^{c_p})}_{\text{alignment term}} + \underbrace{\exp(T - s_\theta^{c_n})}_{\text{constraint term}} \right] \right\} \tag{8}$$

Intuitively, the constraint term increases the score of the negative COT $s_\theta^{c_n}$, with the extent of improvement regulated by the value of $T$. We aim for $T$ to achieve the effect of increasing $s_\theta^{c_n}$ when it is lower than a boundary $B$. In this paper, we chose $B$ as the minimum positive COT score minus a hyper-parameter $\beta$, i.e., $B = s_\theta^{c_p*} - \beta$, where $s_\theta^{c_p*} = \min_{c_p \in \mathbf{G_P}} s_\theta^{c_p}$. To achieve this, we analyze the gradient of Equation 8 with respect to the parameters $\theta$:

$$\nabla_\theta \mathcal{L}_A^{BC} \propto - \sum_{c_p \in \mathbf{G_P}} \sum_{c_n \in \mathbf{G_N}} \left[ \exp(s_\theta^{c_n} - s_\theta^{c_p})(\nabla_\theta s_\theta^{c_p} - \nabla_\theta s_\theta^{c_n}) + \exp(T - s_\theta^{c_n})\nabla_\theta s_\theta^{c_n} \right]$$

$$= - \sum_{c_p \in \mathbf{G_P}} \sum_{c_n \in \mathbf{G_N}} \left\{ \underbrace{\exp(s_\theta^{c_n} - s_\theta^{c_p})\nabla_\theta s_\theta^{c_p}}_{\text{increase } s_\theta^{c_p}} + \underbrace{\left[ \exp(T - s_\theta^{c_n}) - \exp(s_\theta^{c_n} - s_\theta^{c_p}) \right] \nabla_\theta s_\theta^{c_n}}_{\text{change } s_\theta^{c_n} \text{ based on the coefficient}} \right\} \tag{9}$$

Because the score $s_\theta^{c*}$ increases along the gradient $\nabla_\theta s_\theta^{c*}$, based on $\nabla_\theta \mathcal{L}_A^{BC}$, for each pair $(c_p, c_n)$, $\mathcal{L}_A^{BC}$ consistently increases $s_\theta^{c_p}$ due to the positive coefficient $\exp(s_\theta^{c_n} - s_\theta^{c_p}) > 0$. In contrast, it elevates the negative score $s_\theta^{c_n}$ when:

$$\exp(s_\theta^{c_n} - s_\theta^{c_p}) < \exp(T - s_\theta^{c_n}) \Rightarrow s_\theta^{c_n} < \frac{T + s_\theta^{c_p}}{2} = B \tag{10}$$

Otherwise, it tends to decrease or keep the score of $s_\theta^{c_n}$. Combing $B = s_\theta^{c_p*} - \beta$ and Equation 10, we can achieve the value of $T = 2s_\theta^{c_p*} - 2\beta - s_\theta^{c_p}$.

## 4.4 EXTENDING TO RANKING CONSTRAINED ALIGNMENT

The quality of the different COTs is not a simple binary relationship $\mathbf{G_P} \succ \mathbf{G_N}$, i.e., the quality of positive COTs is better than that of negative COTs. In a more general situation, COTs in each

| MODELS | METHODS | GSM8K | AQUA | ECQA | AVERAGE ($\Delta$) |
|--------|---------|-------|------|------|--------------------|
| LLAMA-7B | VFT | 36.48±0.92 | 31.19±0.28 | 70.40±1.07 | 46.02 ( – ) |
| | RFT | 39.75±1.03 | 32.81±1.48 | **72.23±0.11** | 48.28 (↑ 2.26) |
| | AFT ($\mathcal{L}_A^{DC}$) | **40.43±1.04** | 33.01±0.95 | **72.23±0.43** | **48.55 (↑ 2.53)** |
| | AFT ($\mathcal{L}_A^{BC}$) | 40.26±0.36 | **33.20±1.24** | 72.15±0.57 | 48.53 (↑ 2.51) |
| LLAMA2-7B | VFT | 40.71±0.16 | 31.49±1.96 | 72.34±0.22 | 48.18 ( – ) |
| | RFT | 43.65±0.13 | 33.25±1.23 | **73.86±0.38** | 50.25 (↑ 2.07) |
| | AFT ($\mathcal{L}_A^{DC}$) | **44.25±0.43** | **33.49±0.63** | 73.71±0.65 | **50.75 (↑ 2.57)** |
| | AFT ($\mathcal{L}_A^{BC}$) | 44.16±0.81 | 32.89±0.98 | 73.23±0.82 | 50.09 (↑ 1.91) |
| LLAMA-13B | VFT | 42.07±0.15 | 33.91±0.60 | 72.74±0.43 | 49.57 ( – ) |
| | RFT | 46.13±1.41 | 34.29±1.28 | **75.03±0.35** | 51.80 (↑ 2.23) |
| | AFT ($\mathcal{L}_A^{DC}$) | 46.31±1.52 | 34.49±1.21 | 74.32±0.09 | 51.70 (↑ 2.13) |
| | AFT ($\mathcal{L}_A^{BC}$) | **46.46±0.28** | **34.79±0.37** | 74.53±0.68 | **51.93 (↑ 2.36)** |
| LLAMA2-13B | VFT | 47.29±1.24 | 34.68±1.36 | 74.76±0.56 | 52.24 ( – ) |
| | RFT | 50.12±1.57 | 34.95±0.88 | 76.21±0.80 | 53.75 (↑ 1.51) |
| | AFT ($\mathcal{L}_A^{DC}$) | 50.67±1.16 | **35.78±0.45** | 76.42±0.82 | 54.29 (↑ 2.05) |
| | AFT ($\mathcal{L}_A^{BC}$) | **51.03±0.54** | 35.49±1.19 | **76.57±0.83** | **54.36 (↑ 2.12)** |

Table 2: The accuracy of different methods on three reasoning datasets with binary feedback. $\Delta$ denotes the improvement compared to VFT. AFT significantly outperforms VFT and is slightly better than RFT (Yuan et al., 2023a). Note that RFT is a concurrent work to ours.

group can also have quality differences, which means the quality of all generated COTs can be ranked as a sequence $c_1 \succeq c_2 \succeq \cdots \succeq c_k$. If we can obtain such a quality ranking sequence, we can easily extend our binary-feedback boundary-constrained alignment loss $\mathcal{L}_A^{BC}$ to a ranking-feedback boundary-constrained alignment loss as follows:

$$L_A^{RBC} = \log \left\{ 1 + \sum_{c_i \succ c_j} \left[ \underbrace{\exp(s_\theta^{c_j} - s_\theta^{c_i})}_{\text{alignment term}} + \underbrace{\exp(2s_\theta^{c_j*} - 2\beta - s_\theta^{c_i} - s_\theta^{c_j})}_{\text{constraint term}} \right] \right\} \quad (11)$$

Where $s_\theta^{c_j*} = \min_{c_k \succ c_j} s_\theta^{c_k}$ is the minimal score of COTs that have the better quality than $c_j$. Compared with $\mathcal{L}_A^{BC}$, $L_A^{RBC}$ can bring LLMs more detailed training signals of the COT assessment, which can further enhance their performance. We also try to extend $\mathcal{L}_A^{DC}$ to the ranking situation, and we find it slightly underperforms in comparison to $L_A^{RBC}$. Please refer to Appendix D for details.

## 5 EXPERIMENTS

### 5.1 EXPERIMENTAL SETUPS

**Datasets** We conduct our experiments on three widely used reasoning datasets with human-annotated chain-of-thoughts, including math reasoning tasks **GSM8K** (Cobbe et al., 2021), **AQUA-RAT** (Ling et al., 2017) and commonsense reasoning task **ECQA** (Aggarwal et al., 2021). Furthermore, we create **GSM8K-RANK** to evaluate the effectiveness of our AFT in the ranking situation. Please refer to Appendix A for more details of these datasets.

**Parameter Setting** We conduct experiments on four large language models, LLama(2)-7B and LLama(2)-13B. We do not conduct experiments on larger models due to resource limitations. We sample $k = 6$ COTs from VFT-LLMs with a sampling temperature of 1. Our detached constrained alignment loss does not introduce any hyper-parameter, and we search the boundary constraint hyper-parameter $\beta$ based on the validation set. For more training details, please refer to Appendix B.

| METHODS | LLAMA-7B | LLAMA-13B | LLAMA2-7B | LLAMA2-13B | AVERAGE ($\Delta$) |
|---|---|---|---|---|---|
| VFT | 20.82±0.71 | 24.12±0.42 | 24.08±0.22 | 30.28±1.46 | 24.83 ( – ) |
| RFT | 25.09±1.18 | 28.21±0.86 | 28.25±0.78 | 34.53±0.51 | 29.02 (↑ 4.19) |
| RRHF | 7.51±0.56 | 9.92±0.82 | 9.21±0.25 | 13.35±1.26 | 10.00 (↓ 14.8) |
| PRO | 18.73±0.31 | 20.34±1.51 | 21.40±0.92 | 23.55±0.98 | 21.00 (↓ 3.82) |
| AFT ($L_A^{RBC}$) | **26.08±1.05** | **28.97±0.35** | **29.05±0.75** | **35.48±1.35** | **29.90 (↑ 5.07)** |

Table 3: Test accuracy of different methods on GSM8K-RANK with ranking feedback.

**Baselines and Metrics** We compare our AFT with VFT, RFT (Yuan et al., 2023a), RRHF (Yuan et al., 2023b), and PRO (Song et al., 2023a) ( Please refer to Appendix C for details of our baselines). We use the accuracy to measure the model performance. Specifically, we conduct 3 runs with 3 different seeds and report the average results with the standard deviation.

## 5.2 RESULTS WITH BINARY FEEDBACK

Table 2 displays the results of different fine-tuning methods on three reasoning datasets. As is shown: **1)**: AFT significantly outperforms VFT ($p \leq 0.1$, T-test), improving the average accuracy by $1.91\% \sim 2.57\%$ for all models; **2)** Our proposed two constrained alignment strategies slightly outperform RFT with the binary feedback, we think the reason is that the alignment loss can leverage negative examples, which RFT cannot. These results demonstrate the importance of revealing the assessment misalignment problem of VFT and the effectiveness of our AFT approach.

## 5.3 RESULTS WITH RANKING FEEDBACK

As described in Section 4.4, our AFT can also be easily adapted to the ranking situation where we can obtain the quality ranking sequence of generated COTs. Table 3 illustrates the results of different methods in the GSM8k-RANK. As is shown: **1)** Our AFT surpasses all other methods, demonstrating its effectiveness with ranking feedback. For instance, AFT exceeds the strongest baseline RFT by 0.88% in average accuracy. This superiority can be attributed to AFT's ability to help LLMs recognize quality differences among any given pair in a ranking context, while RFT only focuses exclusively on optimizing the probability of the highest-quality examples; **2)** Prior methods utilizing ranking loss have a substantial negative impact on model performance. For example, integrating RRHF loss into VFT leads to a 14.8% reduction in accuracy. In fact, the performance reduction is also observed in their own paper (Song et al., 2023a), which demonstrates that ranking loss often enhances the reward of LLMs, yet results in lower BLEU scores. However, they do not identify the cause, and in this paper, we find that a potential reason for the performance decline is the absence of a constraint in their loss, which we will discuss in Section 6.1.

## 6 ANALYSIS

### 6.1 DELVE DEEPLY INTO RECENT RANKING LOSSES FOR ALIGNMENT

Our experiments on GSM8K-RANK show that adding ranking loss will harm the model performance. We find the reason is that previous alignment ranking losses will unreasonably decrease the score of non-optimal COTs [1]. To empirically validate this hypothesis, we add a detached constraint to these two ranking losses similar to $\mathcal{L}_A^{RDC1}$ (Equation 12). Consequently, these ranking losses will only make the scores of higher-quality COTs larger than those of lower-quality ones, without explicitly decreasing the scores of COTs with lower quality. Table 4 illustrates the final accuracy $\mathbf{T_{Accucary}}$ of different methods in the testing set, the assessment accuracy $\mathbf{A_{Accuracy}}$ and average perplexity of positive COTs $\mathbf{PPL}$ in the training set[2]. As is shown: **1)** Without the constraint strategy, all three ranking losses harm the model performance, leading to higher perplexity and lower final task accuracy compared to VFT; **2)** We observe that the task accuracy of PRO does not decline as significantly as

---

[1]We provide detailed analyses of previous ranking losses for alignment in Appendix F.

[2]For each question in the training set, we sample new COTs (three positive and three negative COTs, respectively) that are different from training COTs for evaluation.

| METHODS | WITHOUT CONSTRAINT | | | WITH CONSTRAINT (OURS) | | |
| --- | --- | --- | --- | --- | --- | --- |
| | $T_{Accuracy}$ | $A_{Accuracy}$ | PPL ($\downarrow$) | $T_{Accuracy}$ | $A_{Accuracy}$ | PPL ($\downarrow$) |
| VFT | **20.82±0.71** | 68.72±1.48 | **1.60±0.01** | 20.82±0.71 | 68.72±1.48 | 1.60±0.01 |
| RRHF | 7.51±0.56 | 87.44±1.28 | 1.80±0.01 | 25.53±0.27 | 79.89±0.60 | **1.35±0.01** |
| PRO | 18.73±0.31 | 86.58±1.09 | 2.34±0.02 | 25.82±0.48 | 80.34±0.97 | 1.45±0.01 |
| AFT ($L_A^{RBC}$) | 7.03±0.98 | **88.89±0.78** | 7.81±0.03 | **26.08±1.05** | **81.36±0.78** | 1.37±0.01 |

Table 4: Task accuracy ($T_{Accuracy}$) and assessment accuracy ($A_{Accuracy}$) on GSM8K for LLama-7B, which is fine-tuned by different methods (with or without constraint) on GSM8K-Rank. PPL ($\downarrow$, lower is better) denotes the average perplexity of all positive COTs.

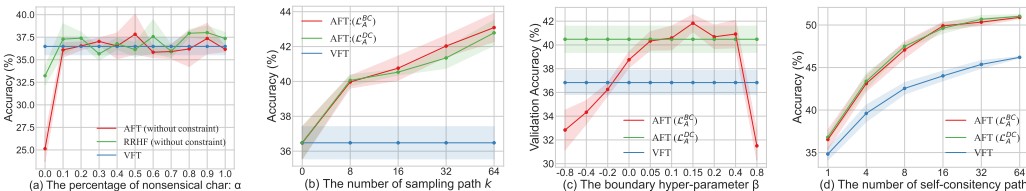

Figure 2: Variation of accuracy with (a) different quality of negative examples; (b): different number of sampling COTs for alignment training; (c) different boundary constraint hyper-parameter; (d) different number of voting paths of self-consistency.

RRHF and AFT. We find the reason is that PRO employs a dynamic temperature that reduces the negative score in a more reasonable manner (Please refer to Appendix F.3 for details); **3)** By adding the constraint, all ranking losses can not only improve two accuracies but also decrease the perplexity. These results show the importance of constraint for other alignment ranking losses. [3]

## 6.2 THE INFLUENCE OF NEGATIVE EXAMPLE QUALITY FOR RANKING LOSSES

In this paper, we attribute the model degradation to the over-punishment of negative examples with good quality. To demonstrate this, we delve deeper into this issue by examining the influence of negative example quality on model performance. To this end, we construct low-quality negative examples by replacing $\alpha$ proportion characters in the original example with nonsensical characters '.' (Please refer to Appendix E for details). We evaluate the performance of three methods, VFT, AFT (without constraint), and RRHF (without constraint), using negative examples across a range of $\alpha$ values from 0 to 1. As shown in Figure 2(a): 1) compared with VFT, two unconstrained ranking losses with high-quality negative examples ($\alpha = 0$) significantly harm the model performance. 2) when the quality of negative examples is low ($\alpha > 0$), the ranking loss generally improves the model performance. These results support our claim that the model's degradation stems from excessively penalizing non-optimal but still good negative examples, and also demonstrate the necessity of constraint in the ranking loss.

## 6.3 EFFECTIVENESS OF THE NUMBER OF CANDIDATE COTS

As described in Section 4.1, AFT samples $k$ candidate generation results to align LLMs. In this section, we explore the influence of $k$. We sampled 0, 8, 16, 32, and 64 results from the VFT-LLama-7B, and then de-duplicated these sampling results. We train LLama-7B on the de-duplicated datasets. As shown in Figure 2(b), we can see that AFT can consistently improve the model performance with $k$ improvement, which is a promising result. We think the reason is that with large $k$, the AFT will have more data to help the LLM perceive the quality of different COT paths, which enhances the final performance. This growing accuracy shows the effectiveness and the potential of AFT.

---

[3] We also conduct a case study in Appendix G to intuitively show the effectiveness of constrained alignment and the model degradation caused by the unconstrained alignment.

| METHODS | GSM8K | AQUA | ECQA | MMLU | AVERAGE ($\Delta$) |
|---|---|---|---|---|---|
| VFT | 35.72$\pm$0.95 | 32.95$\pm$0.98 | 69.25$\pm$0.74 | 37.52$\pm$1.03 | 43.86 ( – ) |
| AFT ($\mathcal{L}_A^{DC}$) | 40.24$\pm$0.63 | 33.72$\pm$0.92 | 71.38$\pm$0.64 | 39.25$\pm$0.35 | 46.15 ($\uparrow$ 2.29) |
| AFT ($\mathcal{L}_A^{BC}$) | 40.00$\pm$0.69 | 33.45$\pm$0.56 | 71.48$\pm$0.89 | 38.89$\pm$0.70 | 45.96 ($\uparrow$ 2.10) |

Table 5: Comparison of VFT- and AFT-LLama-7B with training data "GSM8K+AQUA+ECQA" on three in-domain benchmarks and an out-of-domain benchmark MMLU.

## 6.4 ABLATION ON THE BOUNDARY VALUE

The boundary constraint term of AFT requires a hyper-parameter $\beta$ to regulate the boundary. In this section, we conduct an ablation study to demonstrate the impact of varying $\beta$ values. As depicted in Figure 2(c), the performance initially increases and subsequently decreases as $\beta$ ranges from -0.8 to 0.8. These findings align with expectations, as a small $\beta$ cannot effectively widen the score gap between high-quality and low-quality COTs, while an overly large $\beta$ may result in over-punishment of non-optimal COTs, thereby compromising the model's generative abilities. In conclusion, the results emphasize the importance of the boundary constraint term and indicate that the value of $\beta$ can significantly affect model performance. Therefore, it is essential to carefully adjust this value when using our boundary-constrained alignment loss.

## 6.5 EFFECTIVENESS OF AFT WITH SELF-CONSISTENCY

Self-consistency (Wang et al., 2023c) is a highly effective strategy for improving LLM's reasoning performance. This method involves sampling multiple COTs and utilizing a voting process to determine the final answer during inference. In contrast, AFT samples COTs for training to develop better LLMs. Both methods utilize COTs to enhance the model's reasoning ability, while are utilized at different stages. In this section, we explore the combination of AFT and Self-consistency. As illustrated in Figure 2(d), as the number of paths increases, the improvement of AFT is more significant than VFT, demonstrating that AFT effectively enhances self-consistency. We believe the reason is that AFT helps models learn to assess the quality of different COTs by encouraging larger scores for high-quality COTs compared to low-quality ones. This means that high-quality COTs are more likely to be sampled, and thus, AFT can enhance self-consistency.

## 6.6 EFFECTIVENESS OF AFT ON THE MULTI-TASK AND OUT-OF-DOMAIN SITUATIONS

To further demonstrate the effectiveness and versatility of AFT, we investigate its performance in multi-task scenarios. We combine the training sets of three datasets and use both AFT and VFT to train the LLama-7B model. As depicted in Table 5, AFT is able to simultaneously enhance the performance of all corresponding test sets. Additionally, we evaluate both AFT and VFT on the MMLU (zero-shot), an out-of-distribution benchmark, and AFT also outperforms VFT. These results indicate that AFT not only improves the performance of in-distribution tasks but also enhances the model's transfer ability, leading to significantly better out-of-distribution performance.

## 7 CONCLUSION AND LIMITATIONS

In this paper, we find that the vanilla fine-tuned (VFT) LLMs with chain-of-thought (COT) reasoning process suffer from an assessment misalignment problem, i.e, they fail to access the quality of different COTs of the learned questions, which hinders the reasoning ability of LLMs. To this end, we propose an alignment fine-tuning (AFT) paradigm. Our AFT consists of a novel constrained alignment loss that can align the model assessment behaviors without harming the model performance. Extensive experiments on four reasoning benchmarks with both binary and ranking feedback demonstrate the effectiveness of AFT. In addition, AFT also performs well in multi-task and out-of-distribution situations. Furthermore, we also delve deeply into recent widely used ranking losses for alignment and find that the constraint, which these approaches have overlooked, is also crucial for their performance. Besides the advantages of our work, we also discuss some limitations of our paper. Please refer to Appendix H for details.

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

> I want you to act as a grade school math teacher, and evaluate the quality of the answer provided by an AI assistant to the math Question displayed below.
> You will be given a reference answer and the assistant's answer, and Your evaluation should consider the correctness of the assistant's answer.
> Begin your evaluation by comparing the assistant's answer with the reference answer step-by-step. Identify and correct any mistakes.
> The answer is scored out of 10 points, with one point deducted for each wrong step. Be as objective as possible.
> Your need first provide your Evaluation Evidence and then rate the response on a scale of 1 to 10.
> [Question]:
> {question}
> [The Start of Reference Answer]
> {reference}
> [The End of Reference Answer]
> [The Start of Assistant's Answer]
> {answer}
> [The End of Assistant's Answer]
> You MUST output with two lines:
> Evaluation Evidence: <Explanation>
> Rating: <ONLY a single digit>

Table 6: The evaluation template that prompts ChatGPT to score each candidate COT.

## A  DATASETS

We conduct our experiments on three widely used reasoning datasets with human-annotated chain-of-thoughts, including math reasoning tasks GSM8K (Cobbe et al., 2021), AQUA-RAT (Ling et al., 2017), commonsense reasoning task ECQA (Aggarwal et al., 2021). Furthermore, we create **GSM8K-RANK** to evaluate the effectiveness of our AFT in the ranking situation:

**GSM8K** GSM8K is a widely used mathematical reasoning dataset, which comprises 8.5K varied math word problems for grade school, developed by human authors. It is partitioned into 7.5K training problems and 1K testing problems. We sample 400 problems from the testing set to form the validation set, and thus we have $7,473$, $400$, and $919$ examples in training, validation, and testing sets, respectively.

**AQUA-RAT** AQUA-RAT comprises approximately $100,000$ algebra-based word problems, each accompanied by a natural language rationale. Each example in the dataset consists of four components: 1) question, which statement is written in natural language, 2) options, a set of five potential answers with one being correct, 3) rationale, a natural language explanation of the problem's solution, and 4) correct, the right answer choice. For efficiency, we randomly sample $5,000$, $400$, and $1,254$ examples as the training, validation, and test set, respectively.

**ECQA** ECQA is derived from CommonsenseQA (CQA) (Saha et al., 2018) by generating a free-flow explanation for each QA pair in CQA. CQA is a comprehensive dataset for commonsense reasoning, containing QA pairs with five choices and a single correct answer. ECQA comprises 11K QA pairs in total and has $7,598$, $1,090$, and $2,194$ examples in the training, validation, and test sets, respectively.

**GSM8K-RANK** To evaluate the effectiveness of our AFT in the ranking situation, we randomly select 1,000 examples from GSM8K's training set and generate 8 candidate COTs for each question. We then prompt ChatGPT to rate these candidates by providing the question, reference answer, and the COT to be assessed and thus we can achieve a quality ranking sequence for different generated COTs. We randomly sampled 20 examples and found that ChatGPT's scoring results align well with human assessment. ChatGPT is instructed to assign a score between 1 and 10, indicating the quality of each COT. To ensure the reliability of the ratings, following (Wang et al., 2023b), we require ChatGPT to present evaluation evidence before assigning a score, and simple 3 scores for each example. We take the average score as the final score for each COT.

| Models | GSM8K | AQUA | ECQA | GSM8K-RANK |
|--------|-------|------|------|------------|
| LLama-7B | 0.15 | 0.15 | 0.15 | 0.05 |
| LLama2-7B | 0.15 | 0.40 | 0.35 | 0.15 |
| LLama-13B | 0.15 | 0.15 | 0.15 | 0.15 |
| LLama2-13B | 0.15 | 0.15 | 0.20 | 0.15 |

Table 7: The value of hyper-parameter $\beta$ for boundary Constrained Alignment.

### A.1 WHY WE CHOSE GSM8K, AQUA-RAT AND ECQA AS OUR EVALUATION DATASETS.

In this paper, our objective is to explore the influence of VFT on the reasoning capabilities of LLMs. Currently, mathematics and commonsense reasoning serve as two exemplar tasks for reasoning. As such, we selected our evaluation datasets from these two reasoning task categories. Additionally, we aim to uncover the potential drawback of VFT, which necessitates a chain-of-thought reasoning process for every example in the training set.

The most widely used datasets with chain-of-thought reasoning processes are GSM8K and AQUA-RAT for mathematics and ECQA for commonsense reasoning. Consequently, we opted to use these three datasets as benchmarks for our study. Our research also extends to examining the efficacy of our proposed AFT under ranking circumstances. Due to the absence of appropriate datasets that provide ranking feedback, we developed the GSM8K-RANK dataset.

While there are alternative datasets available, we believe that these three widely used datasets, complemented by our self-constructed GSM8K-RANK, sufficiently demonstrate both the presence of assessment misalignment issues in VFT and the viability of our proposed AFT.

## B  PARAMETER SETTING

We conduct experiments on four large language models, LLama-7B, LLama-13B, LLama2-7B, and LLama2-13B. We do not conduct experiments on larger models due to resource limitations. We sample $k = 6$ COTs from VFT-LLMs with a sampling temperature of 1. Our detached constrained alignment loss does not introduce any hyper-parameters, and we search the hyper-parameter of boundary constraint loss within the range $(0, 0.05, 0.1, 0.15, 0.2, 0.25, 0.3, 0.35, 0.4, 0.45, 0.5)$ on the validation set. The value of $\beta$ of different models and datasets is provided in Table 7. On GSM8K, AQUA, and ECQA, the models are trained for 3, 3, and 1 epochs, respectively. The learning rate is set to 2e-5, featuring linear decay and a linear warmup for 3% of the total training steps. 7B and 13B models are trained on 8 and 32 V100 GPUs with 32GB memory, respectively. We employ a maximum sequence length of 512 and utilize the DeepSpeed library and ZeRO optimizer during training.

## C  BASELINES

We compare our AFT with the following baselines:

**VFT**  the vanilla fine-tuning (VFT) method that simply trains LLMs with the reference COT using the MLE loss, which is the most widely used training strategy.

**RFT**  Rejective sampling fine-tuning (RFT) (Yuan et al., 2023a) selects the COTs with the correct answer, adds these COTs to the origin training data, and uses the new augmented training data to train LLMs, which is proven to be a very strong baseline.

**RRHF**  Rank Responses to align Human Feedback (RRHF) (Yuan et al., 2023b), which takes candidate ranking into account and distinguishes different candidates through a pair-wise ranking loss.

| Methods | $\mathcal{L}_{VFT}$ | $+L_A^{RBC}$ | $+\mathcal{L}_A^{RDC1}$ | $+\mathcal{L}_A^{RDC2}$ | $+\mathcal{L}_A^R$ |
|---|---|---|---|---|---|
| Accuracy | 20.82±0.71 | 26.08±1.05 | 25.68±0.49 | 12.57±1.34 | 7.03±0.98 |

Table 8: Results of LLama-7B on GSM8K fin-tuned by different methods.

| **High-quality** ($\alpha = 0$) | 48 x 2 = << $48 * 2 = 96$ >> 96 clips were sold in April. 48 / 2 = << $48/2 = 24$ >>24 clips were sold in May. So, Natalia sold a total of 96 + 24 = << $96 + 24 = 120$ >>120 clips in April and May. [ANS] 120. |
|---|---|
| **Low-quality** ($\alpha = 0.1$) | $48x.2 = .$ < $48 * 2 = .6$ >> 96 clips were sold in Apri.. $48/2 = .$ < $48/2 = 24$ >> 24 clips were sold in May. So, Natalia sold a total of 96 + 24 = . < $.96 + .4.1... >$ 120 .lips in April and May. [ANS] 120. |

Table 9: High-quality and low-quality negative examples. We construct low-quality negative examples by replacing $\alpha$ of the chars with nonsensical char '.'.

**PRO** Preference Ranking Optimization (PRO) (Song et al., 2023a), which takes candidate ranking into account and distinguishes different candidates through a ranking loss with a dynamic temperature.

# D DETACHED CONSTRAINED RANKING LOSS

Given a ranking sequence $c_1 \succeq c_2 \succeq \cdots \succeq c_k$, besides extending $\mathcal{L}_A^{BC}$ (Equation 8) to the ranking loss $L_A^{RBC}$ (Equation 14), we also try to extend $\mathcal{R}_A^{DC}$ to two types of detached constraint ranking loss as follows:

$$L_A^{RDC1} = \log\left[1 + \sum_{c_i \succ c_j} \exp(\mathbf{D}(s_\theta^{c_j}) - s_\theta^{c_i})\right] \qquad (12)$$

$$L_A^{RDC2} = \log\left[1 + \sum_{c_i \succ c_j, c_j \notin c_{min}} \exp(s_\theta^{c_j} - s_\theta^{c_i}) + \sum_{c_i \succ c_j, c_j \in c_{min}} \exp(\mathbf{D}(s_\theta^{c_j}) - s_\theta^{c_i})\right] \qquad (13)$$

where $c_{min}$ is the set of all lowest-quality examples. Specifically, $L_A^{RDC1}$ detachs the score of $c$ when it serves as a negative example, while $L_A^{RDC2}$ only detach the score of lowest-quality examples. We design $L_A^{RDC2}$ as we consider that in a ranking scenario, higher-quality examples are inherently constrained by lower-quality ones. Consequently, we hypothesize that constraining only the lowest examples could potentially prevent model degradation. We also consider a ranking baseline without any constraint:

$$\mathcal{L}_A^R = \log\left[1 + \sum_{c_i \succ c_j} \exp(s_\theta^{c_j} - s_\theta^{c_i})\right] \qquad (14)$$

Table 8 illustrates the results of LLama7B fine-tuned by different methods on GSM8K-RANK. As is shown: **1**): The method without setting any constraint $\mathcal{L}_A$ only achieves 7.03% accuracy, showing the importance of adding a constraint to the alignment loss. **2**): $L_A^{RDC2}$, which applies a detached constraint solely to the lowest-quality examples, attains a marginally improved accuracy of 12.57%. However, it also considerably impairs the model's overall performance compared with VFT, indicating that constraining only the lowest-quality examples is insufficient. **3**): $L_A^{RDC1}$ is much better than VFT, $L_A^{RDC2}$ and $\mathcal{L}_A$, we think the reason is that after detaching all negative scores, $L_A^{RDC1}$ prevents the model degradation, however, it is worse than $L_A^{RBC}$, we hypnosis that $L_A^{RDC1}$ only tries to improve all scores, although with different extends, which is not good enough in the ranking situation.

# E  THE INFLUENCE OF NEGATIVE EXAMPLE QUALITY

We construct low-quality negative examples by replacing $\alpha$ proportion characters in the original example with nonsensical characters '.'. Intuitively, the larger $\alpha$, the lower the quality of negative examples. Table 9 shows an example of $\alpha = 0$ and $\alpha = 0.1$.

# F  DELVE DEEPLY INTO PREVIOUS RANKING LOSSES FOR ALIGNMENT

In this section, we delve deeply into previous widely used ranking losses for alignment, DPO (Rafailov et al., 2023), RRHF (Yuan et al., 2023b) and PRO (Song et al., 2023a), and point out that they all suffer from lack of a constraint term.

Given a ranking sequence $c_1 \succeq c_2 \succeq \cdots \succeq c_k$, all ranking losses are proposed to ensure the scores of high-quality examples are larger than those of low-quality examples. Ranking losses usually use the token-averaged log-likelihood to represent the score of an example $c$ given by an LLM parameterized by $\theta$:

$$s_\theta^c = \frac{1}{|c|} \sum_{j=1}^{|c|} \log P\left(c_j \mid c_{<j}, q; \theta\right),\qquad(15)$$

## F.1  DPO

Direct Preference Optimization (DPO) (the ranking version) optimizes LLMs with the following ranking loss:

$$
\begin{aligned}
\mathcal{L}_{DPO} &= -\sum_{c_i} \log \frac{\exp(\beta s_\theta^{c_i} - \beta s_{\theta_{ref}}^{c_i})}{\exp(\beta s_\theta^{c_i} - \beta s_{\theta_{ref}}^{c_i}) + \sum_{c_j \prec c_i} \exp(\beta s_\theta^{c_j} - \beta s_{\theta_{ref}}^{c_j})} \\
&= \sum_{c_i} \log \left[ 1 + \sum_{c_j \prec c_i} \exp(\beta s_\theta^{c_j} - \beta s_{\theta_{ref}}^{c_j} - \beta s_\theta^{c_i} + \beta s_{\theta_{ref}}^{c_i}) \right]
\end{aligned}
\qquad(16)
$$

where $\theta$ and $\theta_{ref}$ are parameters of the training model and reference model, respectively. The training model and reference model are usually initialized by the same LLM, and DPO freezes the reference model during fine-tuning. $\beta$ is a hyper-parameter of DPO.

To analyze the effectiveness of DPO, we compute the gradient with respect to the parameters $\theta$:

$$
\nabla_\theta \mathcal{L}_{DPO} = -\sum_{c_i}
$$

$$
\frac{\sum_{c_j \prec c_i} [\beta \exp(\beta s_\theta^{c_j} - \beta s_{\theta_{ref}}^{c_j} - \beta s_\theta^{c_i} + \beta s_{\theta_{ref}}^{c_i}) \nabla_\theta s_\theta^{c_i} - \beta \exp(\beta s_\theta^{c_j} - \beta s_{\theta_{ref}}^{c_j} - \beta s_\theta^{c_i} + \beta s_{\theta_{ref}}^{c_i}) \nabla_\theta s_\theta^{c_j}]}{1 + \sum_{c_j \prec c_i} \exp(\beta s_\theta^{c_j} - \beta s_{\theta_{ref}}^{c_j} - \beta s_\theta^{c_i} + \beta s_{\theta_{ref}}^{c_i})}
$$

$$(17)$$

Based on $\nabla_\theta \mathcal{L}_{DPO}$, for each pair $(c_i, c_j)$, $\mathcal{L}_{DPO}$ will decrease the $s_\theta^{c_j}$ with the gradient weight $\frac{\beta \exp(\beta s_\theta^{c_j} - \beta s_{\theta_{ref}}^{c_j} - \beta s_\theta^{c_i} + \beta s_{\theta_{ref}}^{c_i})}{1 + \sum_{c_j \prec c_i} \exp(\beta s_\theta^{c_j} - \beta s_{\theta_{ref}}^{c_j} - \beta s_\theta^{c_i} + \beta s_{\theta_{ref}}^{c_i})}$, which may lead the model degradation.

In the original DPO paper (Rafailov et al., 2023), they observe this catastrophe and alleviate it by setting a very small $\beta$ (e.g., 0.1) to achieve a small gradient weight. Please refer to the original paper for more details. However, based on Equation 17, the small $\beta$ also hamper the improvement of positive examples, which may also hinder the model's performance. Furthermore, solely relying on reducing gradient weights might not be sufficient to prevent model deterioration, as demonstrated in the subsequent analysis of RRHF and PRO. In this paper, we do not replicate DPO since there is no official public code available for ranking. In addition, compared with other ranking losses, DPO needs an additional reference model, incurring higher memory costs.

| Scaling Factor $\beta$ | 0.1 | 0.2 | 0.3 | 0.4 | 0.5 | 0.6 | 0.7 | 0.8 | 0.9 | 1 |
|---|---|---|---|---|---|---|---|---|---|---|
| Accuracy | 18.75 | 18.01 | 15.05 | 13.20 | 11.79 | 11.79 | 9.83 | 8.78 | 8.62 | 7.51 |

Table 10: The influence of gradient weight scaling factor $\beta$ for RRHF.

## F.2 RRHF

Rank Responses to align Human Feedback (RRHF), which takes candidate ranking into account and distinguishes different candidates through a pair-wise ranking loss:

$$\mathcal{L}_{RRHF} = \sum_{c_i \succ c_j} \max(0, s_\theta^{c_j} - s_\theta^{c_i}) \tag{18}$$

We compute the gradient of $\mathcal{L}_{RRHF}$ with respect to $\theta$:

$$\nabla_\theta \mathcal{L}_{RRHF} = -\sum_{c_i \succ c_j} \left[ \underbrace{\mathbb{I}(s_\theta^{c_j} > s_\theta^{c_i})\nabla_\theta s_\theta^{c_i}}_{\text{increase } s_\theta^{c_i}} - \underbrace{\mathbb{I}(s_\theta^{c_j} > s_\theta^{c_i})\nabla_\theta s_\theta^{c_j}}_{\text{decrease } s_\theta^{c_j}} \right] \tag{19}$$

Based on $\nabla_\theta \mathcal{L}_{RRHF}$, we can see that although RRHF implicitly introduces a constraint by setting the loss to 0 when the positive score is larger than the negative score, it still has a drawback: Whenever $s_\theta^{c_j} > s_\theta^{c_i}$, $\mathcal{L}_{RRHF}$ will decrease the $s_\theta^{c_j}$ with the same gradient weight $\mathbb{I}(s_\theta^{c_j} > s_\theta^{c_i}) = 1$. This weight might be too large, potentially harming the model's performance.

To illustrate this, we explore the performance of RRHF with a scaling factor $\beta$ on its gradient weight. As shown in Table 10, it is evident that as the weight increases (larger $\beta$), the model's performance declines, showing that: 1) The constraint of RRHF is not effective enough to prevent model degradation; 2) We can alleviate the model degradation by making the gradient weight smaller suggested by DPO (Rafailov et al., 2023); 3) Although we have tried a very small $\beta = 0.1$, RRHF still harms the performance, which shows solely relying on reducing gradient weights might not be sufficient to prevent model deterioration.

In fact, in the original RRHF paper (Yuan et al., 2023b), the authors have observed that a large ranking weight, such as 10 or 100, significantly impairs model performance, leading them to try a smaller weight (i.e., 1). However, they do not analyze the potential reason. In this paper, we highlight that a key factor causing this discrepancy is the unwarranted reduction of the negative example score, which necessitates imposing a constraint on the ranking loss. In addition, we discovered that a weight of 1 can also substantially harm the model's performance in the reasoning task. We believe that the optimal weight of RRHF varies across tasks.

## F.3 PRO

Preference Ranking Optimization (PRO), which takes candidate ranking into account and distinguishes different candidates through a ranking loss with a dynamic temperature:

$$\mathcal{L}_{PRO} = -\sum_{c_i} \log \frac{\exp(\tau_i^{max} s_\theta^{c_i})}{\exp(\tau_i^{max} s_\theta^{c_i}) + \sum_{c_j \prec c_i} \exp(\tau_i^j s_\theta^{c_j})} = \sum_{c_i} \log[1 + \sum_{c_j \prec c_i} \exp(\tau_i^j s_\theta^{c_j} - \tau_i^{max} s_\theta^{c_i})] \tag{20}$$

$$\tau_i^j = r^{c_i} - r^{c_j} > 0, \quad \tau_i^{max} = \max_{c_j \prec c_i} \tau_i^j \tag{21}$$

where $r^c$ is the score of $c$ given by a reward model. $\tau_i^j$ is the dynamic temperature for score $s_\theta^{c_j}$. We compute the gradient with respect to the parameters $\theta$:

$$\nabla_\theta \mathcal{L}_{PRO} = -\sum_{c_i} \frac{\sum_{c_j \prec c_i} [\tau_i^{max} \exp(\tau_i^j s_\theta^{c_j} - \tau_i^{max} s_\theta^{c_i})\nabla_\theta s_\theta^{c_i} - \tau_i^j \exp(\tau_i^j s_\theta^{c_j} - \tau_i^{max} s_\theta^{c_i})\nabla_\theta s_\theta^{c_j}]}{1 + \sum_{c_j \prec c_i} \exp(\tau_i^j s_\theta^{c_j} - \tau_i^{max} s_\theta^{c_i})} \tag{22}$$

| Methods | PRO | PRO (remove $\tau$) | PRO + RDC1 | PRO (remove $\tau$) + RDC1 |
|---------|-----|---------------------|------------|----------------------------|
| Accuracy | 18.73±0.31 | 7.18±0.78 | 25.84±0.48 | 25.43±0.98 |

Table 11: The importance of dynamic temperature of PRO. "remove $\tau$" denotes remove the dynamic temperature term, i.e., $\tau_j^j$ and $\tau_i^{max}$ from PRO. "+RDC1" denotes add our ranking detach technical (Equation 12).

Based on $\nabla_\theta \mathcal{L}_{PRO}$, we can see that for each pair $(c_i, c_j)$, $\mathcal{L}_{PRO}$ will decrease $s_\theta^{c_j}$ with the dynamic gradient weight:

$$\text{DGW}_i^j = \frac{\tau_i^j \exp(\tau_i^j s_\theta^{c_j} - \tau_i^{max} s_\theta^{c_i})}{1 + \sum_{c_j \prec c_i} \exp(\tau_i^j s_\theta^{c_j} - \tau_i^{max} s_\theta^{c_i})}, \tag{23}$$

which may harm the model's performance. However, the dynamic gradient weight that is computed based on the reward is more reasonable than the constant value of 1 used in RRHF, and thus PRO outperforms RRHF. Specifically, when there is a substantial reward gap between higher-quality and lower-quality, indicated by a large value $\tau_i^j$. It is reasonable to increase the penalty for negative example scores (large $\text{DGW}_i^j$), and vice versa. To demonstrate this, we remove the dynamic temperature term, i.e., $\tau_j^j$ and $\tau_i^{max}$, from PRO. As shown in Table 11, we can see that PRO significantly outperforms PRO (remove $\tau$) when there is no constraint. However, the performance gap shrinks when adding our detached constraint. These results indicate: 1) To a certain extent, the dynamic temperature's effectiveness stems from its ability to make PRO reduce the negative score in a more reasonable manner. 2) The dynamic temperature is useful to prevent model degradation but is not good enough.

### F.4 SUMMARY

Our analysis reveals that previous ranking-based alignment works have some limitations:

1) Although their methods consist of some strategies to prevent model degradation (i.e., using a scaling factor to reduce gradient weight for DPO, setting the loss to 0 for RRHF, and incorporating a dynamic temperature for PRO), they do not recognize the importance of constraints for ranking-based alignment methods in their papers.

2) Their strategies essentially involve diminishing the gradient weight's magnitude, which is proven to be insufficient (at least in the reasoning tasks).

Different from previous works, in this paper:

1) We point out an assessment misalignment problem of VFT for reasoning and highlight the importance of constraint for alignment to prevent model degradation.

2) We introduce a novel constrained alignment loss. The constrained alignment loss with a boundary constraint term not only alters the magnitude but also adjusts the direction of the gradient weight depending on the condition, which is proven to be very effective in preventing model degradation and enhancing the reasoning ability of LLMs.

## G CASE STUDY

We also conducted a case study to intuitively show the importance of our constrained alignment. As shown in Table 12, given the question, our AFT successfully gives the correct COT and answer, while VFT gives the wrong COT at the second step (colored red), demonstrating the superiority of AFT. More importantly, after removing the boundary constraint, the generative ability of LLM seems to degrade, resulting in outputting many repeat and meaningless output tokens.

## H LIMITATIONS

Our paper has some limitations, which should be discussed in future works: **1)** Due to the resource limit, we do not scale the AFT to larger LLMs such as 65B and 70B LLama models. However, we

| Question | *At the end of the school year, Kate asked her teacher if she could have the 3 boxes of 64 crayons since they were all worn down to small pieces. The teacher agreed. Kate took the crayons home and decided to make crayon muffins by melting 8 small pieces of crayons together in a muffin tin that she placed in an oven for a few minutes. If she can sell her muffin crayons for $1.50 each, how much money can Kate make?* |
|---|---|
| **Answer** | She was able to get 3 boxes of 64 crayons so she has 3*64 = <<3*64=192>>192 crayons Each crayon muffin needs 8 crayons and she has 192 crayons so she can make 192/8 = <<192/8=24>>24 crayon muffins If she makes 24 crayon muffins and sells them for $1.50 apiece, she can make 24*1.5 = $<<24*1.5=36.00>>36.00 [ANS] 36. |
| **VFT** | She has 3 boxes of crayons that are all worn down to small pieces so she has 3*64 = <<3*64=192>>192 pieces of crayons She melts 8 pieces of crayons together in a muffin tin so she has 8+192 = <<8+192=200>>200 crayon muffins She can sell her muffin crayons for $1.50 each so she can make 200*1.5 = $<<200*1.5=300>>300 [ANS] 300. |
| **AFT** | Kate has 3 boxes * 64 crayons = <<3*64=192>>192 crayons. She melts 192 crayons / 8 crayons per muffin = <<192/8=24>>24 muffins. She sells the muffins for $1.50 each and has 24 muffins, so she makes 24 muffins * $1.50/muffin = $<<24*1.5=36>>36. [ANS] 36. |
| **AFT** (*remove constraint*) | She can make 8 pieces of crayons per muffin tin so 3 boxes * 8 pieces/tin = 8 * 8 = <<3*8=24>>24 pieces fit in 1 tin Each melted crayon is worth $1.50 because 24 pieces * 1.50/64 pieces = 1.50/1.50/24 pieces = 1.50/1.50 She can make 4 tins of melted crayons because 24 pieces / 1.50 / 1.5 / 24 pieces / 1.50 / 1.50 / 24 pieces / 1.50 / 1.50 / 24 pieces / 1.50 / 1.50 / 24 pieces / 1.50 / 1.50 / 24 pieces / 1.50 / 1.50 / 24 pieces / 1.50 / 1.50 / 24 pieces / 1.50 / 1.50 |

Table 12: A case study to intuitively show the effectiveness of AFT with boundary constraint. the right and wrong steps are colored by blue and red, respectively.

believe that larger models still suffer from the assessment misalignment problem of VFT, and thus AFT can improve the performance of these larger models; **2)** Our boundary-constrained alignment loss incorporates a hyper-parameter $\beta$ that regulates the constraint strength, significantly impacting the model's performance. Finding the optimal hyper-parameter requires constructing a validation set and a certain search overhead. Although our detached alignment loss can mitigate the assessment misalignment problem without requiring any hyper-parameters, it sometimes falls short in comparison to the boundary-constrained alignment loss, especially in ranking situations. Therefore, how to design a dynamic boundary constraint without introducing the hyper-parameter is a meaningful question, which leaves for further work.

