# OpenReview forum: "Making Large Language Models Better Reasoners with Alignment"
_ICLR.cc/2024/Conference — Submitted to ICLR 2024_

### Official Review · Reviewer_9VoV · 2023-10-17

**Soundness:** 3 good
**Presentation:** 3 good
**Contribution:** 2 fair
**Rating:** 5
**Confidence:** 4

**Summary:**

The paper addresses reasoning problems using LLMs and Chain-of-Thought (CoT).
The paper proposes to sample multiple chains of thought of the same training question from a pretrained model, and finetune the model to prefer the solutions that lead to the correct final answer. This results in improvements on several reasoning benchmarks, compared to the baseline which was only finetuned on the training set without this augmentation.

**Strengths:**

* The proposed approach is simple
* The paper focuses on a class of important problems
* The approach results in gains across multiple popular benchmarks

**Weaknesses:**

1. The proposed approach is very similar to [LARGE LANGUAGE MODELS CAN SELF-IMPROVE (Huang et al., 2022)](https://arxiv.org/pdf/2210.11610.pdf), which came out a year ago. Since the authors did not cite it, I assume that they were not aware of it, but in terms of novelty there is a significant overlap.

2. Motivation - The motivation in Table 1 is unclear. T-Accuracy is ~40% but ~A-Accuracy is ~70% - Is it a surprising result?
The paper says that:
>These results show that the assessment ability of VFT-LLMs is far from expected, as they cannot
accurately discern the quality of various COTs of previously learned questions.

I'm not sure I agree. What other results would the authors expect?

3. Over-mathematical - I think that there are large complicated parts in the paper that are not necessarily needed, and the paper can be significantly simplified. Since "Detached Constraint" (Section 4.3.1) and "Boundary Constraint" (Section 4.3.2) perform almost the same, while none of them consistently outperforms the other, why do we need both of them?

**Questions:**

### Questions
1. The paper says that:
>We discover that LLMs fine-tuned by the vanilla fine-tuning ... frequently assign lower scores to high-quality COTs compared to low-quality ones

Which is correct, but isn't it trivial? Isn't it the case with any machine learning model - sometimes the model assigns higher probability to the wrong output and low probability to the correct output? Isn't this the source of any kind of mistake in any machine learning model?

### Comments
1. While terms such as "serve as the brain of the artificial general intelligence" (appearing twice) are unfortunately popular in media, have no scientific basis, and I suggest avoiding them in a research paper.
2. Figure 1 is confusing, or there is a mistake in the text that refers to it:  the second paragraph of the Introduction says:

>As a result, they struggle to assess the quality of other answers and tend to assign lower perplexity (higher score) to
incorrect Candidate Answer 1 compared to the correct Candidate Answers 2.

However, Answer 1 **is the correct answer**, and Answer 2 is the incorrect.

3. There are some claims that are inaccurate. For example:
> Intuitively, the MLE objective seeks to exclusively allocate probability mass to the reference COT

I wouldn't say that it *exclusively* allocates probability mass to the reference COT, since a lot of mass remains for other possible CoT. As evidence, their probability is not zero.

As another example:
>As demonstrated by our pilot experiment, VFT-LLMs fail to give reasonable scores to COTs in GP and GN.

What are "reasonable scores"? What scores did the authors expect?

4. Figure 2 is visually nice, important, and extensive, but unfortunately impossible to read because the fonts are too tiny.
5.  The experiments were performed across multiple benchmarks (which is great), using the 7B and 13B versions of LLama 1 and 2. However, I think that these models were only pretrained, without instruction tuning or RLHF. It would be great if the authors could also experiment with the "Chat" version of Llama 2 (of the same sizes).

### Summary
I appreciate the authors' efforts and extensive analysis, but I think that the main approach is too similar to a previous work that came out a year ago (and was not cited). This fact severely hurts the paper in terms of novelty. I thus vote for rejection at this time, unless convinced that there is a significant difference that I have missed.

---

> ### Author Response · Authors · 2023-11-16
> **Responese to your comment (1/2)**
>
> Thank you for your valuable review comments. However, there are **some misunderstandings** regarding both the **novelty** and **motivation** of our work, we hope our explanation and responses can resolve your misunderstanding and concerns.
>
> ***Q1: Question about the novelty: the proposed approach is very similar to [Huang et al., 2022)](https://arxiv.org/pdf/2210.11610.pdf) [1], which severely hurts the paper in terms of novelty.***
>
> A1: It is essential to point out that the contributions between [1] and our work are **totally different.**
>
> [1] aim to achieve self-improvement of LLMs with two steps:
>
> 1. Using LLM to generate solutions for a given question with chain-of-thought promoting and select high-confidence solutions with self-consistency techniques.
> 2. Fine-tuning the LLMs on the selected self-generated solutions with a next-token prediction objective to achieve self-improvement.
>
> **Totally Different** from [1]:
>
> 1. We discover that LLMs fine-tuned by the vanilla next-token prediction objective **suffer** from an assessment misalignment problem.
> 2. We propose an alignment fine-tuning (AFT) paradigm to address our identified problem. Our proposed AFT method consists of novel and effective alignment loss to train LLMs, which is totally different from the next-token prediction objective used in [1].
> 3. We delve deeply into recent ranking-based alignment methods (which are truly similar works to ours) and discover that the constraint, which has been overlooked by these approaches, is also crucial for their performance.
>
> Given such **obvious differences in the contributions** between [1] and our work. We can confirm that the contributions of these two works are orthogonal.
>
> ***Q2: Question about the motivation: The results of our pilot experiment (Table 1) are trivial and not surprising. What are the expected results of our pilot experiment, and what is the motivation derived from the pilot experiment?***
>
> A2: The results of our pilot experiment inspired us to propose an effective alignment fine-tuning paradigm to improve the reasoning ability of LLMs, which is not trivial for the following reasons:
>
> In this paper, we define two important abilities of LLM reasoners: 1) **reasoning ability**: LLMs can generate the right solution **for the unlearned question**; 2) **assessment ability**: LLMs can evaluate the quality of different candidate solutions **for the learned question**, i.e, they can assign higher scores (lower perplexity) to right solutions compared with wrong solutions.
>
> Previous studies have shown that the vanilla chain-of-thought fine-tuning (VFT) can significantly improve the reasoning abilities of LLMs. However, is VFT sufficient for teaching LLM reasoning ability? We conducted a pilot experiment to explore this question.
>
> If VFT is sufficient enough, we **expect that LLMs trained with VFT will demonstrate strong assessment ability**. This is based on a reasonable assumption that LLMs that truly learn to solve a reasoning question should be able to assess the quality of different candidate solutions to the learned question.
>
> Contrary to our expectations, our pilot experiments show that VFT-LLMs lack this desired assessment ability: the assessment accuracy on **their learned questions** is just around 70% on GSM8K and 62% on ECQA.
>
> **Their assessment accuracy is surprising (NOT trivial) and far from our expectations because**:
>
> 1. the assessment task in our pilot experiments is a binary classification task where a random selection baseline can have up to 50% assessment accuracy.
> 2. The questions of our pilot assessment task are from the training set, and LLMs have already been trained on these questions.
> 3. With the right training in problem-solving, we as humans can achieve near-perfect assessment accuracy. Take Figure 1 as an example, after learning to generate the reference solution for the given question, it becomes very easy for humans to evaluate the quality of two candidate solutions for the learned question. However, this level of assessment is currently beyond the capabilities of VFT-LLMs.
>
> In addition, the pilot experiment also shows that **the reasoning ability on unlearned questions has a strong positive correlation to the assessment ability of learned questions**.
>
> Therefore, our pilot experiences inspire us to improve the reasoning ability of LLMs by proposing an assessment alignment loss to improve their assessment and reasoning ability.

---

> > ### Author Response · Authors · 2023-11-16
> > **Responese to your comment (2/2)**
> >
> > ***Q3: What are "reasonable scores"? What scores did the authors expect? (Question to our statement “As demonstrated by our pilot experiment, VFT-LLMs fail to give reasonable scores to chain-of-thought (COTs) in GP and GN.”)***
> >
> > A3: The “reasonable scores” imply that VFT-LLMs should give lower perplexity (high score) to right COTs in GP compared to wrong COTs in GN for their learned questions. However, our pilot experiments reveal that VFT-LLMs fail to give such “reasonable scores”, i.e., they have poor assessment ability. In addition, our pilot experiment also revealed that the assessment ability has a strong correlation to the final reasoning ability. Therefore, we propose an alignment fine-tuning paradigm to enhance their assessment ability to improve the reasoning ability of LLMs.
> >
> > ***Q4: Since "Detached Constraint (DC)" (Section 4.3.1) and "Boundary Constraint (BC)" (Section 4.3.2) to perform almost the same, while none of them consistently outperform the other, why do we need both of them?***
> >
> > A4: Both BC and DC offer unique benefits. DC stands out for its simplicity and lack of additional hyper-parameters, making it easy to implement. BC introduces a boundary hyper-parameter and generally presents better results. For instance, Figure 2(b) shows that as the number of training solutions increases, BC's performance advantage over DC becomes more pronounced. Furthermore, Figure 2(c) reveals that by adjusting the boundary hyper-parameter, BC can achieve an almost 2% increase in accuracy compared to DC on the validation set, highlighting BC's potential.
> >
> > ***Q5: Experiment results on the Chat version of Llama 2.***
> >
> > A5: As you suggest, we conduct further experiments on the Chat version of LLM, i.e., LLama-2-7B-chat and LLama2-13B-chat on GSM8K:
> >
> > | Methods (on 7B-Chat) | Accuracy | Methods (on 13B-Chat) | Accuracy |
> > | --- | --- | --- | --- |
> > | VFT | 39.03 | VFT | 47.19 |
> > | AFT (DC) | 43.82 | AFT (DC) | 50.32 |
> > | AFT (BC) | 44.04 | AFT (BC) | 51.01 |
> >
> > As is shown, our AFT significantly outperforms VFT with the Chat version models as the backbone.
> >
> > [1] Large Language Models Can Self-Improve. arXiv:2210.11610

---

> > ### Comment · Reviewer_9VoV · 2023-11-20
> > **Response to authors**
> >
> > Thank you for your response.
> >
> > The authors write:
> > >Totally Different from [1]:
> > We discover that LLMs fine-tuned by the vanilla next-token prediction objective suffer from an assessment misalignment problem.
> >
> > As I asked in my review: "isn't it trivial? Isn't it the case with any machine learning model - sometimes the model assigns higher probability to the wrong output and low probability to the correct output? Isn't this the source of any kind of mistake in any machine learning model?"
> >
> > I don't see this discovery as a significant contribution.
> >
> > The authors write:
> > >We propose an alignment fine-tuning (AFT) paradigm to address our identified problem. Our proposed AFT method consists of novel and effective alignment loss to train LLMs, which is totally different from the next-token prediction objective used in [1].
> >
> > The question is whether this additional loss is really needed.
> > The fact that the loss proposed in this work is "different" is not sufficient; maybe the part of generating various CoT solutions (which as far as I understand, is common to the two papers) is the key, and then training on these generated solutions can be done even with the standard next-token loss?
> >
> > Since no comparison was made to the Huang et al. paper, I as a reader cannot tell whether the additional loss is really needed.
> >
> > The authors write:
> > >Given such obvious differences in the contributions between [1] and our work. We can confirm that the contributions of these two works are orthogonal.
> >
> > Since the paper does not make any conceptual or empirical comparison to [1] (Huang et al.,), I cannot fully agree with this statement.
> >
> > The authors write:
> > >If VFT is sufficient enough, we expect that LLMs trained with VFT will demonstrate strong assessment ability. This is based on a reasonable assumption that LLMs that truly learn to solve a reasoning question should be able to assess the quality of different candidate solutions to the learned question.
> >
> > I find it hard to agree with this assumption. There is [growing evidence](https://arxiv.org/pdf/2310.01798.pdf) in the literature. that solving a reasoning question and assessing the quality of different candidates are not always the same thing and not always correlated.
> >
> > The authors write:
> > >Contrary to our expectations, our pilot experiments show that VFT-LLMs lack this desired assessment ability: the assessment accuracy on their learned questions is just around 70% on GSM8K and 62% on ECQA.
> >
> > As I asked in my review, why are these numbers surprising? The number "70%" on its own is meaningless without comparison to any alternative (that is non-human and non-random). Did the authors expect 100%? What about 80% or 90%, are they close to expected or not?
> >
> > The authors write:
> > >the assessment task in our pilot experiments is a binary classification task where a random selection baseline can have up to 50% assessment accuracy.
> > ...
> > With the right training in problem-solving, we as humans can achieve near-perfect assessment accuracy.
> >
> > The authors are saying that a random baseline achieves 50% accuracy and a human achieves 100% accuracy. It sounds only reasonable to me that an LLM would achieve something between 50-100%.

---

> ### Author Response · Authors · 2023-11-18
> **Looking Forward to Your Reply!**
>
> Dear Reviewer 9VoV,
>
> Thank you again for your valuable feedback and comments! We would greatly appreciate it if you could let us know whether you are satisfied with our response, especially the response about the **novelty** and **motivation** of our paper. We will be happy to address any remaining concerns.
>
> Sincerely, Paper7016 Authors

---

> ### Author Response · Authors · 2023-11-19
> **Looking Forward to Your Reply!**
>
> Dear Reviewer 9VoV,
>
> Thank you again for your valuable feedback and comments!
>
> We can confirm that there exist some misunderstandings regarding both the novelty and motivation of our work.
>
> We greatly need and look forward to discussions with you to eliminate misunderstandings.
>
> We would also be delighted to address any other concerns.
>
> Sincerely, Paper7016 Authors

---

> ### Comment · Reviewer_9VoV · 2023-11-20
> **Response to authors (2/2)**
>
> The authors write:
> >A3: The “reasonable scores” imply that VFT-LLMs should give lower perplexity (high score) to right COTs in GP compared to wrong COTs in GN for their learned questions. However, our pilot experiments reveal that VFT-LLMs fail to give such “reasonable scores”, i.e., they have poor assessment ability
>
> It feels that the authors refer to relative properties with absolute and over-decisive conclusions.
> The authors write that "VFT-LLMs _fail_ to give reasonable scores"; however 70% doesn't sound like a complete failure to me, and it doesn't sound like a "poor assessment ability" if the only comparison is to a human who can achieve 100%.
>
> >As you suggest, we conduct further experiments on the Chat version of LLM, i.e., LLama-2-7B-chat and LLama2-13B-chat on GSM8K
>
> Thank you for these additional experiments, I think they are an easy experiment that strengthens the paper with more empirical evidence.
>
> ## Conclusion
>
> Although our main discussion here focused on the question of whether "70% assessment is high or low?", I think this issue can be easily solved by toning down this part.
>
> The main concern that I have is the lack of conceptual and empirical comparison to [Huang et al., (2022)](https://arxiv.org/pdf/2210.11610.pdf). I am not one of the authors of that paper, and it doesn't bother me personally that this paper was not cited. I honestly want to understand the difference and whether the simpler approach of Huang et al. may be sufficient, without introducing additional losses.

---

> ### Author Response · Authors · 2023-11-21
> **Responese to your comment**
>
> Dear Reviewer 9VoV:
>
> Thank you again for your valuable feedback and comments! Please see the responses as follows:
>
> ***Q1: Regarding the empirical comparison to Huang et al., (2022). — Your main concern***
>
> A1: The review writes:
> ```
> The main concern that I have is the lack of conceptual and empirical comparison to Huang et al., (2022).
> ```
> ```
> The fact that the loss proposed in this work is "different" is not sufficient; maybe the part of generating various CoT solutions (which as far as I understand, is common to the two papers) is the key, and then training on these generated solutions can be done even with the standard next-token loss?
> ```
>
> I agree with you that generating various CoT solutions is important, and training LLMs on these generated solutions with a standard next-token loss is a necessary and strong baseline. **It is essential to point out that we have compared our AFT with such baseline, i.e., Rejective Sample Fine-tuning (RFT, a concurrent work to ours)**. RFT further trains LLMs on the high-quality generated solutions with the correct answer using the next token prediction. **Our experiments have demonstrated that AFT outperforms RFT, especially in the ranking situation.**
>
> **Although Huang et al., (2022) focus on the self-improvement of LLMs on unlabeled datasets, which is totally different from ours**, we have also followed your suggestion to replicate Huang et al., (2022)'s approach. This replication involves training LLama-7B on their generated CoTs with the majority voting answer.
>
> | Methods | GSM8K | GSM8K-RANK |
> | --- | --- | --- |
> | VFT | 36.48 | 20.82 |
> | Huang et al., (2022) | 38.23 | 22.34 |
> | RFT | 39.75 | 25.09 |
> | AFT | 40.26 | 26.08 |
>
> As is shown, our AFT outperforms Huang et al., (2022).  **More importantly, from the principles of the methods, AFT can better utilize the feedback compared with RFT or Huang et al., (2022).** Specifically, AFT has the ability to help LLMs recognize quality differences among any given COT pair in a ranking context, while the RFT or Huang et al., (2022)'s approach can not achieve this because they only try to optimize the probability of the chosen COTs.
>
> Based on the experimental results and the underlying principles of these methods, it's clear that AFT is essential. We hope these results can address your main concern.
>
> ***Q2: About the assessment ability for VFT-LLMs.***
>
> The reviewer writes:
> ```
> sometimes the model assigns higher probability to the wrong output and low probability to the correct output? Isn't this the source of any kind of mistake in any machine learning model? isn't it trivial?
> ```
> It is trivial for the model to assign a higher probability to the wrong output for their **unlearned question**. However, we believe doing so for their **learned questions** is not trivial. If we think VFT is good enough to teach LLM reasoning, it will be really surprising when VFT-LLMs stumble on extremely easy assessment tasks for their learned questions, such as Figure 1.
>
> The reviewer writes:
>
> ```
> Why are these numbers surprising? The number "70%" on its own is meaningless without comparison to any alternative (that is non-human and non-random). Did the authors expect 100%? What about 80% or 90%, are they close to expected or not?
> ```
>
> In Table 4, we have compared the assessment ability of VFT with that of our AFT (a non-human and non-random alternative). We include the results here for your convenience:
>
> | Methods | Reasoning Accuracy | Assessment Accuracy |
> | --- | --- | --- |
> | VFT | 20.82 | 68.72 |
> | AFT | 26.08 | 81.36 |
>
> Please note that the COTs used to measure the assessment ability here are newly sampled COTs, distinct from the COTs used to train AFT.
>
> This result demonstrates that there is substantial room for improvement in the assessment ability of VFT, further affirming the validity of our motivation: **Enhancing the reasoning ability of LLMs by addressing the assessment misalignment problem in VFT.**
>
> We would greatly appreciate it if you could let us know whether you are satisfied with our response. We will be happy to address any remaining concerns.

---

> > ### Comment · Reviewer_9VoV · 2023-11-21
> > **Response to authors**
> >
> > Thank you for your response.
> >
> > I still think that the question of whether the model's assessment accuracy "is high or low" is not really meaningful. It's a nice anecdote to show that it improves, but it's not really indicative on its own, and I'm not sure that great conclusions should be made from the absolute number of "70%".
> >
> > Regarding RFT - thank you for clarifying that the RFT baseline is similar to the Huang et al (2022) paper that I mentioned.
> > I recommend mentioning Huang et al (2022) in the paper, and explaining the difference between Huang et al and this paper, and whether Huang et al is different from RFT.
> >
> > I increased my score to 5. I am still concerned that the improvement over RFT is minor.
> > The authors write that:
> > >AFT has the ability to help LLMs recognize quality differences among any given COT pair in a ranking context, while the RFT or Huang et al., (2022)'s approach can not achieve this
> >
> > theoretically speaking, yes, but in practice, the improvement is even lower than the standard deviation, while RFT is much simpler and does not require additional losses.

---

> ### Author Response · Authors · 2023-11-22
> **Response to your comment**
>
> Thank you again for your valuable feedback and comments. Please see our responses as follows:
>
> ***Q: Is our conclusion made from an absolute number?***
>
> The reviewer writes:
>
> ```
> I still think that the question of whether the model's assessment accuracy "is high or low" is not really meaningful. It's a nice anecdote to show that it improves, but it's not really indicative on its own, and I'm not sure that great conclusions should be made from the absolute number of "70%".
> ```
>
> **The number 70% is not only an absolute number,** it symbolizes a lot of surprising instances like Figure 1. The VFT-LLMs fail to give reasonable scores in such frustratingly easy assessment tasks for their learned question. If you think such **stupid mistakes on their ***LEARNED*** question** are trivial, let's leave aside the debate about this. At least, these results can fully demonstrate that VFT is insufficient to cultivate a real reasoner.
>
> **Our conclusion is not made from an absolute number.**  Our pilot experiment aims to point out that VFT is not enough, and thus we need a new fine-tuning paradigm.
> How to design such a fine-tuning paradigm? Our pilot experiment reveals a strong positive correlation between assessment and reasoning accuracy (with Pearson Correlation Coefficients of 0.93 and 0.98 at GSM8K and ECQA, respectively). **Therefore, the main conclusion in our pilot experiment is that we can improve the reasoning ability of LLMs by improving their assessment ability. The conclusion is derived from the correlation, not an absolute number.** In addition, our experiments have confirmed our conclusion.
>
>
> ***Q: Regarding the advantages of AFT.***
>
> The reviewer writes:
>
> ```text
> Theoretically speaking, yes, but in practice, the improvement is even lower than the standard deviation, while RFT is much simpler and does not require additional losses.
> ```
>
> **As previously agreed upon, in theory, AFT boasts a distinct advantage over RFT** because of its ability to leverage ranking feedback, an attribute that RFT lacks. Beyond theoretical considerations, **our empirical experiments have demonstrated the potential of AFT**. Compared with binary feedback, the performance gap between AFT and RFT becomes more pronounced when using ranking feedback. **The expanding performance gap demonstrates the theoretical advantage of AFT**.
>
> We concur with your assessment that RFT is a straightforward and effective strategy. Nonetheless, our AFT also boasts ease of implementation.
>
> **The theoretical advantage, practical experiments, and the simplicity of the implementation collectively underscore the necessity of AFT.**
>
> We will add a comparison with Huang et al (2022) in our paper, and we hope that our response can persuade you to buy our idea. Thank you for your consideration.

---

### Official Review · Reviewer_MzKF · 2023-11-01

**Soundness:** 3 good
**Presentation:** 3 good
**Contribution:** 3 good
**Rating:** 6
**Confidence:** 4

**Summary:**

This work identified an Assessment Misalignment problem in pre-trained Large Language Models (LLMs), where these models cannot well distinguish subpar Chain of Thought (COT) reasoning processes from good COT reasoning processes. The paper then proposed an Alignment Fine-Tuning (AFT) paradigm to address this Assessment Misalignment problem. AFT addresses this by a three-step process: fine-tuning LLMs with COT data, generating multiple COT responses per question, and calibrating the scores using their proposed constraint alignment loss. The AFT method is validated through extensive experiments, showing improved performance in reasoning tasks across various benchmarks​.

====After authors' discussion===
I have read through the authors' response, and I think they have addressed my concerns. Therefore, I keep my score that this is a work marginally above the acceptance threshold.

**Strengths:**

[+] The paper identified an important problem that may be overlooked in existing literature -- the misaligned assessment on different COT reasoning process

[+] The proposed method achieved empirical improvement over vanilla finetuning and other baselines on several datasets

**Weaknesses:**

[-] The improvements over existing methods seem a little bit incremental.

[-] see questions

**Questions:**

- It would be great if the authors could provide some intuitions on their designed losses to address the corresponding constraint
- It would be great if the authors could explain why the performance drop for other baseline methods when comparing to vanilla finetuning
- I also wonder how the quality of LLM-generated COTs impact the performance of AFT. For example, how large is the variance using 3 generated examples?

---

> ### Author Response · Authors · 2023-11-16
>
> Thank you for your valuable review comments.
>
> ***Q1: It would be great if the authors could provide some intuitions on their designed losses to address the corresponding constraint.***
>
> A1: In this paper, we find that the rank-based alignment loss without constraint will lead to the collapse of LLMs. We point out this is because the unconstraint alignment loss will over-punish the negative solutions that also express some reasonable reasoning process (Our analyses in Sections 6.1 and 6.2 substantiate this conclusion). Therefore, we design two constant alignment losses, detached constraint alignment losses (DC) and boundary constraint alignment losses (BC) to align LLMs without over-punishing the negative solutions.
>
> ***Q2: It would be great if the authors could explain why the performance dropped for other baseline methods when compared to vanilla fine-tuning.***
>
> A2: As illustrated in Sections 6.1 and 6.2, the performance of other ranking-based methods drops because they lack constraint in their losses and over-punish negative COTs. After adding our constraint, they outperform the vanilla fine-tuning.
>
> ***Q3: I also wonder how the quality of LLM-generated COTs impacts the performance of AFT. For example, how large is the variance using 3 generated examples?***
>
> A3: Our AFT enhances the reasoning capabilities of LLMs by improving their assessment ability on sampled candidate COTs. We discovered that the diversity of these sampled candidates is crucial. For instance, we found that data sampled from models fine-tuned over two epochs outperforms data sampled from models fine-tuned over three epochs, with a noteworthy accuracy difference approaching 1%.
>
> On your suggestion, we sampled 3 groups of generated COTs (each group comprising 3 COTs per question) on GSM8K from the fine-tuned Llama-7B model. We then used this sample data to further fine-tune the Llama-7B model using AFT. The standard deviations measured were 0.42 and 0.53 for AFT-DC and AFT-BC, respectively. The results appear stable, which we believe can be attributed to the homogeneity in the quality of samples drawn from the same model.
>
> The question of how to define quality, and how to sample high-quality solutions from fine-tuned LLMs, remains a valuable area of exploration, which we leave for future work.

---

### Official Review · Reviewer_rVEU · 2023-11-01

**Soundness:** 2 fair
**Presentation:** 3 good
**Contribution:** 2 fair
**Rating:** 5
**Confidence:** 4

**Summary:**

The paper proposes an improved fine-tuning procedure for LLMs to keep high chain of thought reasoning capabilities. The authors therefore propose a constrained alignment loss based on a constrastive loss function and constraints for the gradients of negative examples. The approach is evaluated on three reasoning datasets - GSM8K, AQUARAT, ECQA and a self-created extension of GSM8K. The chosen baselines are RFT, RRHF, PRO and vanilla fine-tuning. The results are on-par or superior to the baselines.

**Strengths:**

The authors propose a sensible approach to do fine-tuning. The proposed fine-tuning loss including the constraints for negative examples is sufficiently introduced and defined. The method is also easily applicable to other problems, given that negative samples are identified. Also, the authors provide runnable code for the review, backing up the clarity and quality of their work.

The evaluation results are promising as well. The approach is mostly better than the chosen baselines, thereby showing improved reasoning capabilites. Here, the chosen baselines are quite sensible, as they include one approach tailored for mathematical reasoning (RFT) as well as general fine-tuning results (RRHF, PRO). Given the larger related work, it remains open what the current SoTA results are.

In a similar vein, it is quite clear from the paper where the loss design differences to the baselines of the evaluation lie, but originality wrt to some referenced works is more difficult to assess from the paper alone.

**Weaknesses:**

The related work for preference alignment a tad vague: Although it includes the a variety of strongly related and relevant works, the focus of the discussion could/should be more on the diverse strategies of the LLMs tuned for mathematical reasoning tasks. Referenced works could thus be better introduced and compared to based on the respective losses/techniques. This would make clear how innovative/novel the proposed technique is.

There is no clear argumentation why other mathematical datasets are not used /or referenced in order to back up the design decision for the chosen datasets. It would be good/important to introduce a clear argumentation or reference why these datasets have been chosen, as there are other/more datasets in this field.

There is no evaluation against some of the direct competitors, such as the referenced Li et al., 2023. It would important to argument why these models have not been chosen for comparison - maybe it is not required. Otherwise it is difficult to understand for the reader if the proposed approach supersedes the current State-of-the-Art. As the approach of the paper can be applied to other/general fine-tuning problems, the added value could also be shown by comparing on more general datasets.

**Questions:**

Did you compare your methods to other approaches focussed on chain-of-though reasoning for mathematical tasks?

Why are the chosen evaluation datasets sufficient for your claims? Are these the main datasets of other related works in the field or other reasoning datasets "easier" than the chosen ones?

Are the empirical results on-par with other referenced works in the field, such as Li et al., 2023?

How would standard RLHF perform here? It would be an interesting baseline, as no constrains on the ranking loss are put and it is simpler than PRO.

How difficult is it to set hyperparameter $B$ and what implications does it have on the results?

---

> ### Author Response · Authors · 2023-11-16
> **Responese to your comment (1/2)**
>
> Thank you for your valuable review comments.
>
> ***Q1: Did you compare your methods to other chain-of-though reasoning approaches or other fine-tuning strategies for mathematical tasks?***
>
> A1: Our work is orthogonal to existing chain-of-thought reasoning approaches in mathematical tasks, which typically fall into two categories: chain-of-thought prompting and chain-of-thought fine-tuning.
>
> 1. Chain-of-thought prompting, which is orthogonal to our work, focuses on triggering the reasoning capabilities of Language Models (LLMs) without tuning them. It complements our proposed Alignment Fine-Tuning (AFT) methods. Our analysis, illustrated in Figure 2(d), investigates the integration of our AFT with Self-Consistency [1], a prevalent chain-of-thought prompting ensemble strategy, and demonstrates that AFT and Self-Consistency complement each other.
> 2. Chain-of-thought fine-tuning, which primarily involves collating chain-of-thought data (such as with Flan [2], Distill-Step-by-step [3], MetaMATH [4], MAmmoTH [5]) and training LLMs with standard next-token prediction objective. Our work highlights the limitations of this vanilla chain-of-thought fine-tuning (VFT) paradigm and introduces an alternative AFT method. Our experiments demonstrated AFT outperforms VFT.
>
> To the best of our knowledge, VFT is the most widely used and effective fine-tuning strategy to improve the reasoning ability of  LLMs [2] [3] [4] [5], and **we are the first work to point out the assessment misalignment problem of VFT for reasoning, and we proposed AFT to address this problem**. Addressing the assessment misalignment problem is a new perspective to improve the reasoning ability of LLMs. There are no known exploration or fine-tuning strategies aimed at addressing this problem specifically within the context of mathematical tasks. Therefore, the most relevant approaches to ours are other alignment fine-tuning methods, such as RRHF and PRO. In this paper, we delve deeply into these methods, and our proposed constraint can also enhance these alignment-tuning methods.
>
> We will enrich our paper by incorporating more discussion related to mathematical tasks within the section dedicated to related work.
>
> ***Q2: Why are the chosen evaluation datasets sufficient for your claims? Are these the main datasets of other related works in the field or are other reasoning datasets "easier" than the chosen ones?***
>
> A2: We focus on two typical reasoning tasks, including mathematical reasoning and commonsense reasoning. For the mathematical reasoning task, we employ GSM8K and AQUA-RAT as our benchmarks, while we use ECQA for the commonsense reasoning task. All of these benchmarks are broadly recognized and used in relevant research works [1] [2] [3] [4] [5] [6] [7].
>
> We observe the assessment misalignment problem of VFT on both mathematical and commonsense reasoning tasks, and our proposed AFT significantly outperforms VFT on all benchmarks. These results are sufficient for our claims.
>
> ***Q3: Are the empirical results on-par with other referenced works in the field, such as Li et al., 2023?***
>
> A3: Li et al., 2023 [6] is a chain-of-thought prompting work, which is orthogonal work to ours. They find that better reasoning performance of LLMs can be achieved by a step voting strategy.
>
> Our work focuses on revealing the assessment misalignment problem of the vanilla chain-of-thought fine-tuning (VFT) paradigm, and proposes an alignment fine-tuning (AFT) paradigm to address the find problem. Therefore, we choose VFT and recent alignment methods as our baselines.  Our experiments demonstrate the importance of assessment misalignment problems and the effectiveness of our AFT.

---

> > ### Author Response · Authors · 2023-11-16
> > **Responese to your comment (2/2)**
> >
> > ***Q4: How would standard RLHF perform here? It would be an interesting baseline, as no constraints on the ranking loss are put and it is simpler than PRO.***
> >
> > A4: Using standard Reinforcement Learning algorithms such as Proximal Policy Optimization (PPO) requires a well-trained Reward Model which needs careful human annotation and data quality control, as well as significantly more computational resources, which is beyond the scope of this paper and left for further work.
> >
> > In addition, our proposed AFT is a ranking-based preference alignment fine-tuning method, and the most relevant baselines of our work are other ranking-based methods such as RRHF and PRO that are analyzed in our paper.
> >
> > ***Q5: How difficult is it to set hyper-parameters B and what implications does it have on the results?***
> >
> > A5: The process of selecting the boundary hyper-parameter B can be efficiently conducted using a validation set, which is outlined in Appendix B. The value of B is very important for the model performance. As shown in Figure 2(c), the performance initially increases and subsequently decreases as B increases. These findings align with expectations, as a small B cannot effectively widen the score gap between high-quality and low-quality COTs, while an overly large B may result in over-punishment of non-optimal COTs, thereby compromising the model’s generative abilities.
> >
> > [1] Self-Consistency Improves Chain of Thought Reasoning in Language Models. ICLR2023.
> >
> > [2] Scaling Instruction-Finetuned Language Models. arXiv:2210.11416.
> >
> > [3] Distilling Step-by-Step! Outperforming Larger Language Models with Less Training Data and Smaller Model Sizes. ACL2023
> >
> > [4] MetaMath: Bootstrap Your Own Mathematical Questions for Large Language Models. arxiv:2309.12284
> >
> > [5] MAmmoTH: Building Math Generalist Models through Hybrid Instruction Tuning. arXiv:2309.05653
> >
> > [6] Making Large Language Models Better Reasoners with Step-Aware Verifier. ACL2023
> >
> > [7] ...

---

> ### Author Response · Authors · 2023-11-18
> **Looking Forward to Your Reply!**
>
> Dear Reviewer rVEU,
>
> Thank you again for your valuable feedback and comments! We would greatly appreciate it if you could let us know whether you are satisfied with our response. We will be happy to address any remaining concerns.
>
> Sincerely, Paper7016 Authors

---

> > ### Comment · Reviewer_rVEU · 2023-11-19
> > **Thank you for the clarifications**
> >
> > I thank the authors for their responses to my questions.
> >
> > Wrt Q1/Q3: While your and the mentioned approaches are orthogonal, can one subsume from the empirical resutls which direction is more fruitful? Also, would your approach then improve chain-of-though reasoning approaches and did you experiment in that direction?
> >
> > Wrt Q2: It would be nice to see a referenced argumentation in the paper why the approaches were chosen. Do other relevant competitors use this exact setup? There might be other mathematical reasoning datasets used in the evaluation of relevant competitors and while it might be infeasible to evaluate on all fitting datasets, it would be important to understand (from the argumentation in the paper) why the evaluation is relevant as is.
> >
> > Wrt Q4: I think there is a misunderstand, I was referring to standard RLHF, not established RL algorithms.

---

> ### Author Response · Authors · 2023-11-19
> **Responese to your valuable feedback and comments.**
>
> Dear Reviewer rVEU,
>
> Thank you again for your valuable feedback and comments! Please see the responses as follows:
>
> ## Response to WrtQ1/Q3
>
> It's not straightforward to discern which direction yields more fruit as each path offers substantial and valuable insights. However, **our experiments demonstrate that our proposed AFT can improve the chain-of-thought reasoning approaches in other directions**:
>
> 1. For chain-of-thought prompting direction approaches, which aim to trigger the internal reasoning ability of LLMs. Our AFT actually improves the upper bound of these prompting approaches, because AFT can enhance the internal reasoning ability of LLMs.  For example, as shown in Figure 2(d),  our AFT can strengthen self-consistency, which is a prominent strategy in the chain-of-thought prompting approach.
> 2. For chain-of-thought fine-tuning direction approaches, which are fundamentally data-centric because they focus on gathering high-quality training datasets with the chain-of-thought process. All these works utilize VFT to train LLMs. As shown in our main results (Table 2), our AFT can enhance these data-centric approaches by better utilizing their gathered data compared with VFT.
>
> ## Response to Wrt Q2
>
> ***Q: Why do we choose GSM8K, AQUA-RAT, and ECQA as our evaluation datasets?***
>
> A: As you suggest, we add an explanation in the paper.  Please refer to Appendix A.1. for details.
>
> ***Q: Do other relevant competitors use this exact setup?***
>
> A: It may be more accurate to use **relevant works** instead of **relevant competitors,**  because **we are the first work to point out there exists an assessment misalignment problem in VFT, which proposes a new perspective to improve the reasoning ability of LLMs**. The relationship between our work and relevant works is not competitive, because they are orthogonal and complementary.
>
> We do not utilize an identical setup to our relevant works. Indeed, currently, there is no uniformity in the setups of other related works either. For instance, the training data, pre-trained LLMs, and data processing strategies employed in each work are nearly always distinct. In addition, different from previous works, our aim does not lie in achieving state-of-the-art results in mathematics or commonsense reasoning tasks. Rather, our primary focus is on identifying, analyzing, and mitigating the previously unidentified assessment misalignment problem of VFT.
>
> ## Response to Wrt Q4
>
> We have some confusion about the “standard RLHF”.  In our opinion (as also described in our related work, Section 2.2), there exist three typical kinds of preference alignment methods that can alleviate our revealed assessment misalignment problem: (1) reinforcement learning; (2) rejective sampling; and (3) ranking losses.
>
> Does the "standard RLHF" denote the “reinforcement learning” methods? In this paper, we do not analyze this type of method, because it is well known that the reinforcement learning methods are very difficult to tune and need much more computation resources. More importantly, our AFT belongs to ranking losses, and thus we have devoted a lot of space to analyzing other ranking losses.
>
> We would greatly appreciate it if you could let us know whether you are satisfied with our response. We will be happy to address any remaining concerns.

---

> > ### Comment · Reviewer_rVEU · 2023-11-20
> > **Thank you for you further elaborations**
> >
> > Thanks again for your answers!
> >
> > To Q4: I was referring to a basline preferenced-based RL approach [1], which can be seen as simplified version of PRO. As it is often used as baseline in RL from human heedback, it is an interesting baseline.
> >
> > [1] Christiano, P.F., Leike, J., Brown, T., Martic, M., Legg, S. and Amodei, D., 2017. Deep reinforcement learning from human preferences. Advances in neural information processing systems, 30.

---

> > > ### Author Response · Authors · 2023-11-22
> > > **Responese to your valuable feedback and comments.**
> > >
> > > ***Q: Regarding the Standard RLHF baseline***
> > >
> > > A: Thanks for your valuable comments. The standard RLHF method used in your mentioned paper, i.e., TRPO,  also requires a well-trained reward model and needs more computational resources compared with the ranking-loss methods. We chose to reproduce PPO over TRPO for our experiments because of PPO's wider usage in RL-based methods. Specifically, the policy, reward, critic, and reference models are all LLama-7B. We use 32 A100-40G to reproduce the RLHF baseline (Note that the ranking loss just needs 8 A100-40G).
> > >
> > > | Methods | Accuracy |
> > > | --- | --- |
> > > | VFT | 20.82 |
> > > | PPO | 21.32 |
> > > | AFT | 26.08 |
> > >
> > > Our comparative analysis demonstrates that while PPO underperforms compared to AFT, it does surpass VFT. This is notable, especially given our extensive efforts in tuning PPO. Furthermore, PPO incorporates essential constraints like the KL-Divergence term and entropy bonus, which share a similar effect to AFT on model training. These not only prevent the overfitting of negative examples but are also critical for stable training in RL, as highlighted in numerous studies including [1]. Our experiments removing these constraints from PPO led to a model collapse, similar to issues observed with unconstraint ranking losses.
> > >
> > > We would greatly appreciate it if you could let us know whether you are satisfied with our response. We will be happy to address any remaining concerns.
> > >
> > > [1] ****Understanding the impact of entropy on policy optimization, ICML****

---

### Official Review · Reviewer_voWS · 2023-11-01

**Soundness:** 3 good
**Presentation:** 3 good
**Contribution:** 3 good
**Rating:** 6
**Confidence:** 4

**Summary:**

The authors propose a method to improve the chain-of-thought reasoning training by adding a loss function that imposes additional constraints such that sampled generated outputs that reach the correct answer are consistently favored over those with incorrect answer.
The method is evaluated on several reasoning datasets and is shown to outperform existing methods.

**Strengths:**

Overall the paper is easy to read and the presentation of the main ideas is clear.

The proposed method seems novel and is well-motivated. The empirical results are convincing.

**Weaknesses:**

Although the intention is to improve the "reasoning" capability of the model, the additional loss function makes use of the slightly risky assumption that generated outputs with the correct final answer should be assigned higher score than those with the wrong final answer. One  could argue that the chain of thoughts itself is perhaps more important than the final answer and some negative examples should still be scored higher than positive examples with "wrong" reasoning steps. Obviously this cannot be done without additional annotation and the proposed approach seems to work fine despite the risk.

As in label smoothing, one wonders whether a simple entropy penalty can already help improve the "overly high confidence" problem in the first place.

**Questions:**

See above.

---

> ### Author Response · Authors · 2023-11-16
>
> ***Q1: The slightly risky assumption that generated outputs with the correct final answer should be assigned a higher score than those with the wrong final answer.***
>
> A1: Thank you for your valuable review comments. In this paper, we manually check 50 pairs of incorrect-correct pairs, discovering that in 48 out of these 50 pairs, the solution with the correct answer surpassed the one with the incorrect answer in terms of quality.  In addition, our proposed alignment fine-tuning (AFT) paradigm significantly outperforms the vanilla chain-of-thought fine-tuning (VFT) paradigm. These evidences substantiate that it is reasonable to consider solutions with correct answers superior to those with incorrect ones.
>
> Moreover, the main contribution of our paper does not reside in the development of the feedback signal. Rather, it fundamentally revolves around pinpointing the assessment misalignment problem within VFT and introducing AFT as a solution to this issue. We believe that with more precise feedback, our AFT will yield more significant improvement.

---

> > ### Comment · Reviewer_voWS · 2023-11-22
> >
> > Thanks, I maintain my rating.

---

### Meta-Review · Area_Chair_fGMt · 2023-12-05

**Metareview:**

The paper introduces the Alignment Fine-Tuning (AFT) for LLMs for math reasoning tasks using model-generated CoTs.
Specifically, AFT use a modification of the InfoNCE loss where we constraint the score of the negative CoTs (that do not lead to the right answer) reasonable range. This constraint is empirically shown to be crucial for AFT as well as other ranking-based alignment approaches to work for reasoning tasks. The paper presents a neat idea and is well written.


The primary weakness of the paper is that AFT results in marginal gains (that do not seem to be statistically significant) over RFT / ReST, a much simpler approach that only fine-tunes on model-generated CoTs that lead to the correct answer. As such, the empirical benefits of AFT does not justify the implementation complexity and additional hyperparameters introduced by AFT. An interesting comparison to run would be when we scale the number of model-generated COTs, how does RFT compares to AFT (in principle AFT can use all the CoTs while RFT can only use the correct CoTs).  Additionally, some reviewers pointed to writing issues and claims that were not well justified that were unaddressed by the authors.

**Justification For Why Not Higher Score:**

Proposed approach leads to marginal gains over a much simpler fine-tuning approach on model-generated CoTs as well as some unjustified claims that remain unaddressed.

**Justification For Why Not Lower Score:**

N/A

---

### Decision · Program_Chairs · 2024-01-16

Reject